# Economic evaluation of point-of-care testing and treatment for sexually transmitted and genital infections in pregnancy in low- and middle-income countries: A systematic review

Olga P. M. Saweri[1,2]*, Neha Batura[3], Rabiah Al Adawiyah[1], Louise M. Causer[1], William S. Pomat[2], Andrew J. Vallely[1,2], Virginia Wiseman[1,4]

**1** The Kirby Institute, University of New South Wales, Sydney, Australia, **2** The Papua New Guinea Institute of Medical Research, Goroka, Papua New Guinea, **3** University College London, London, United Kingdom, **4** London School of Hygiene and Tropical Medicine, London, United Kingdom

* nsaweri@kirby.unsw.edu.au

## Abstract

### Background

Sexually transmitted and genital infections in pregnancy are associated with adverse pregnancy and birth outcomes. Point-of-care tests for these infections facilitate testing and treatment in a single antenatal clinic visit and may reduce the risk of adverse outcomes. Successful implementation and scale-up depends on understanding comparative effectiveness of such programmes and their comparative costs and cost effectiveness. This systematic review synthesises and appraises evidence from economic evaluations of point-of-care testing and treatment for sexually transmitted and genital infections among pregnant women in low- and middle-income countries.

### Methods

Medline, Embase and Web of Science databases were comprehensively searched using pre-determined criteria. Additional literature was identified by searching Google Scholar and the bibliographies of all included studies. Economic evaluations were eligible if they were set in low- and middle-income countries and assessed antenatal point-of-care testing and treatment for syphilis, chlamydia, gonorrhoea, trichomoniasis, and/or bacterial vaginosis. Studies were analysed using narrative synthesis. Methodological and reporting standards were assessed using two published checklists.

### Results

Sixteen economic evaluations were included in this review; ten based in Africa, three in Latin and South America and three were cross-continent comparisons. Fifteen studies assessed point-of-care testing and treatment for syphilis, while one evaluated chlamydia. Key drivers of cost and cost-effectiveness included disease prevalence; test, treatment, and staff costs; test sensitivity and specificity; and screening and treatment coverage. All studies

**Data Availability Statement:** All relevant data are within the manuscript and its Supporting Information files.

**Funding:** WANTAIM is a partnership of academic and governmental institutions in Papua New Guinea, Australia and Europe. The trial is funded by a Joint Global Health Trials award from the UK Department for International Development, the UK Medical Research Council and the Wellcome Trust (MR/N006089/1); a Project Grant from the Australian National Health and Medical Research Council (GNT1084429); and a Research for Development award from the Swiss National Science Foundation (IZ07Z0_160909/1). OPMS and RA are supported by The University of New South Wales Scientia Higher Degree Candidate Scholarship Scheme. All authors' affiliated institutions contributed via facilities and/or salary contributions. The funders had no role in study design, data collection and analysis, decision to publish, or preparation of the manuscript.

**Competing interests:** The authors have no competing interests.

**Abbreviations:** BV, bacterial vaginosis; CHEERS, consolidated health economic evaluation reporting standard; CT, chlamydia; DALYs, disability-adjusted-life-years; dRDT, dual rapid diagnostic test for HIV and syphilis; dRST, dual rapid syphilis test to determine Treponema and non-Treponema infections; HIV, human immunodeficiency virus; hRDT, point-of-care test for HIV; IEC, Information/ Education/ Communication; LMIC, low- and middle- income countries; MDA, mass drug administration; NG, gonorrhoea; PRISMA, Preferred Reporting Items for Systematic Reviews and Meta-Analyses; RPR, repaid plasma regain test; RST, rapid syphilis test; STI, sexually transmitted and genital infection; TPHA test, Treponema pallidum haemagglutination assay test; TV, Trichomoniasis.

met 75% or more of the criteria of the Drummond Checklist and 60% of the Consolidated Health Economics Evaluation Reporting Standards.

## Conclusions

Generally, point-of-care testing and treatment was cost-effective compared to no screening, syndromic management, and laboratory-based testing. Future economic evaluations should consider other common infections, and their lifetime impact on mothers and babies. Complementary affordability and equity analyses would strengthen the case for greater investment in antenatal point-of-care testing and treatment for sexually transmitted and genital infections.

## Introduction

Sexually transmitted and genital infections (henceforth, referred to as STIs) during pregnancy are associated with a number of adverse pregnancy and birth outcomes [1–9] and their burden is highest in low- and middle- income countries (LMICs) [10–12]. Among the most prevalent infections are the curable STIs: syphilis, gonorrhoea (NG), chlamydia (CT), trichomoniasis (TV) and bacterial vaginosis (BV) [13, 14]. Untreated STIs in pregnancy can be associated with miscarriage, pre-term birth, stillbirth, low birth weight and neonatal eye and respiratory infections [15, 16].

Evidence shows that early detection and treatment of HIV and syphilis during pregnancy reduces the risk of adverse pregnancy and birth outcomes [17–20]. The World Health Organization (WHO) currently recommends HIV and syphilis testing for all pregnant women attending antenatal clinics [21–23]. The recommendation is driven by the commitment to eliminate the mother-to-child-transmission of HIV and syphilis [24]. Effective screening programs in antenatal clinics therefore play a pertinent role in ensuring early detection and treatment of HIV and syphilis, which directly improve maternal and child health.

Interventions utilised for diagnosing and/or treating STIs are illustrated in Table 1. For more than two decades, the diagnosis of STIs in many LMIC settings has been based on the WHO-endorsed strategy of syndromic management i.e. clinical diagnosis with no laboratory confirmation [25]. Syndromic management is often inaccurate and misses asymptomatic infections that make up a significant proportion of STIs in women [26, 27]. Laboratory-based

**Table 1. Common STI interventions to detect and treat sexually transmitted and genital infections in low- and middle- income countries.**

| Intervention | Definition |
| --- | --- |
| Syndromic management | Identification of signs and symptoms associated with STIs and commencing treatment to alleviate symptoms and treat the infection [32]. |
| Laboratory-based testing | Diagnosing STIs by determining the etiological agents responsible for the current infection. Testing requires skilled personnel and controlled conditions specific to a laboratory setting. Results may not be available to the clinician until several days later, requiring patients to return for the results and treatment at a later date [33]. |
| Point-of-care testing | Diagnosing STIs by determining the etiological agent responsible for the current infection at the time of the initial patient consultation. Specimen transport is minimised or not required. Minimal training is required to perform the test. Testing may be done onsite in-front of or near to the patient. Patients should ideally receive results and treatment prior to leaving the clinic [34]. |

diagnosis is beyond the reach of many health services in LMICs due to technical requirements and costs and even where it is available, delays in testing and the provision of results often prevent the timely initiation of treatment [28–31].

Technological advancements and the drive to find diagnostic solutions suitable for use at point-of-care have led to the development of a number of accurate, portable, simple-to-use and low- cost tests that are reshaping the global landscape of STI diagnosis and management [35]. These include rapid, point-of-care tests for HIV and syphilis that have been adopted and scaled up in many antenatal clinic settings and more recently, molecular assays for the diagnosis of CT, NG, and TV [36–38]. These new tests hold considerable promise for LMICs [39–41].

As with any health technology, it is crucial to consider any clinical benefits of point-of-care testing and treatment for STIs in pregnancy along with the associated costs of implementation and scale-up. This is especially important for LMICs that need to prioritise investment across a range of diagnostic technologies, treatments, and diseases within a relatively small fiscal space [42]. For these countries, there is considerable interest in weighing up the potential savings associated with the rapid delivery of results, reduced loss-to-follow-up and reduction of facility costs versus the long-run benefits associated with a laboratory confirmed diagnosis [43]. A new landscape for diagnostics is emerging in LMICs [37, 41, 44] and understanding the resource implications [45] and equity impact [46–48] on the health system and patients is a priority for policymakers [49]. To date, there have been no systematic reviews of the economic evidence relating to point-of-care testing and treatment for STIs in pregnancy. Consolidation and appraisal of studies in this field is timely and necessary for formulating strategies to achieve Sustainable Development Goal (SDG) 3: ensuring good health and well-being [50]. SDG 3 includes ending preventable deaths of newborns and children under 5 years of age and ensuring universal access to sexual and reproductive healthcare services [50, 51]. This systematic review examines economic evaluations of point-of-care testing and treatment of the most burdensome, curable STIs in pregnancy in LMICs. The specific objectives of this review are to:

1. Identify and synthesise the evidence from economic evaluations of point-of-care testing and treatment for STIs in pregnancy in LMICs;

2. Compare and contrast the findings, including key drivers of costs and cost-effectiveness; and

3. Appraise methodological and reporting quality using the Drummond 10 point checklist [52] and the Consolidated Health Economics Evaluation Reporting Standards (CHEERS) checklist [53].

## Materials and methods

The methods for this review follow the Preferred Reporting Items for Systematic Reviews and Meta-Analyses (PRISMA) guidelines (S1 File). The methodology summarised below adheres to the published systematic review protocol [54]. The review is registered in PROSPERO (CRD42018109072).

### Literature search and study selection

A comprehensive literature search was conducted by two researchers (OPMS and NB) in MEDLINE, Embase, and Web of Science and completed in April 2020. The search terms, shown in Table 2, were developed with the help of medical librarians to ensure a sensitive search specific to the objectives of the systematic review. All database searches were identical. Keywords and MeSH terms framed the searches, while truncation was used to capture multiple

**Table 2. Search terms used to identify economic evaluations of point-of-care testing and treatment for STIs in pregnancy in LMIC.**

| Sub-heading search terms | Search terms |
|---|---|
| Economic Evaluations | Cost-Benefit Analysis/ |
| | (cost effectiveness or cost benefit analysis or cost utility or cost analysis).mp. |
| Point-of-Care testing and treatment | Point-of-Care Testing/ |
| | ("point of care" or "rapid" or "bedside" or "near to patient" or "lateral flow" or "test*" or "screening").mp. |
| STIs | GONORRHEA/ |
| | exp Syphilis/ |
| | exp Trichomoniasis/ |
| | exp Chlamydia/ |
| | bacterial vaginosis.mp. |
| | exp "bacterial vaginosis"/ |
| | (STI or STD or "sexual transmitted disease*" or "sexual* transmitted infection*").mp. |
| Pregnancy | ("pregnancy" or "pregnant women" or "ANC" or "antenatal").mp. |

terms and the Boolean operators "OR" and "AND" to combine sub-heading search terms. Thereafter, a simplified version of the search was conducted in Google Scholar and restricted to the first 100 citations. Finally, a hand search of the bibliographies of the articles selected for full text review was conducted.

Two researchers independently conducted the database searches, screened records, titles and abstracts using Microsoft Excel (version 365), read full texts, and hand searched references of the included articles (OPMS and NB). The full texts were independently assessed against the eligibility criteria. A third senior researcher resolved any disagreements (VW).

Studies were included in this review if they:

- conducted a full economic evaluation (comparing the costs and consequences of two or more options and include cost effectiveness analyses (CEAs), cost utility analyses (CUAs), cost-benefit analyses, or cost consequence analyses [55]) or a partial economic evaluation (measuring program or disease costs without comparisons with alternative options or outcomes [55]) of a point-of-care testing and treatment intervention for syphilis, NG, CT, TV or BV;

- focused on pregnant women;

- took place in at least one LMIC, as defined by the World Bank [56]; and

- were full papers published in a peer-reviewed journal (commentaries, conference abstracts, editorials, protocols, and review papers were excluded).

No publication date nor language filter was applied.

## Data extraction and analysis

Data were extracted into a form developed, using Microsoft Excel (version 365), specifically for this review. Variables extracted were guided by the categories in the CHEERS checklist and included: study setting; type of STI; type of economic evaluation; time horizon; type of testing and treatment intervention and comparator; study perspective; types of costs; measures of effectiveness and cost-effectiveness; and sensitivity analysis results.

Data extraction highlighted the significant methodological heterogeneity between studies in terms of interventions, study design, cost categories, and health outcomes. Consequently,

neither a meta-analysis nor a sub-group meta-analysis was able to be performed as originally envisaged [54]. The analysis for this review was limited to a descriptive summary and narrative synthesis. This entailed the tabulation of study characteristics and a comprehensive assessment of relevant themes [57]. In addition, percentage differences were calculated to demonstrate a unitless relative difference and describe how cost-effective an intervention is. The percentage difference is calculated by finding the absolute difference in cost-effectiveness ratios dividing them by the average of the two values and multiplying this by 100. Percentage differences alongside cost-effectiveness decision rules and key outcomes emphasize between study heterogeneity. No additional statistical analyses were performed.

### Study appraisal

Two checklists were used to appraise the methodological quality and reporting standards of the studies included in this review. Methodological quality was assessed using the 10 point, 13-criteria Drummond checklist [52] and reporting quality was appraised using the CHEERS checklist [53]. Together the checklists ensure reporting transparency and consistency of appraisal of the studies included in this systematic review and optimised their comparability across common themes. Each item on the checklist was assigned 'Yes', 'No' or 'Unclear' and where the checklist item was not applicable, 'N/A', was used. All studies included in this systematic review were independently appraised by two researchers (OPMS and NB). A third, senior, researcher (VW) resolved any disagreements.

## Results

### Search results

A total of 532 studies were identified after the initial search of the electronic databases. Sixteen studies fulfilled the inclusion criteria. The selection strategy is illustrated by the PRISMA flow diagram in Fig 1.

Tables 3 and 4 summarise the key results of this systematic review. All studies assessed antenatal point-of-care testing and treatment for syphilis [58–70], except for one that focused on CT [71]. Two studies that assessed syphilis also evaluated point-of-care testing and treatment for HIV in pregnant women [72, 73]. Most studies were conducted in Africa (n = 10) [58, 59, 61–64, 66, 70, 71, 73], while three were conducted in Latin and South America [67, 68, 72] and the remaining three involved cross-country comparisons [60, 65, 69]. Of the single country analyses, four countries were classified as low-income countries [63, 68, 70, 73] and seven as middle-income countries [59, 62, 64, 66, 67, 71, 72]. For the remaining multi-country studies, the majority of countries were middle-income countries [58, 60, 61, 65, 69].

### Economic evaluations

**Type of economic evaluation.**   Economic evaluations were categorised as either partial or full evaluations Table 3 indicates that four studies were partial economic evaluations [60, 63, 64, 72]. The remaining 12 studies were full economic evaluations, which included two cost-consequence analyses [61, 73] and 10 CEAs [58, 59, 62, 65–71], seven of which were CUAs [58, 65–70].

**Perspectives, costs, and outcomes.**   Nine studies conducted an economic evaluation using trial data [59, 60, 63, 64, 66–69, 72], while seven studies utilised existing literature [58, 61, 62, 65, 70, 71, 73]. Ten studies took a provider perspective [58–60, 63–66, 69, 71, 72], three took a societal perspective [61, 68, 73] and three studies did not specify the perspective taken [62, 67, 70]. The most common types of costs measured were direct and indirect medical costs

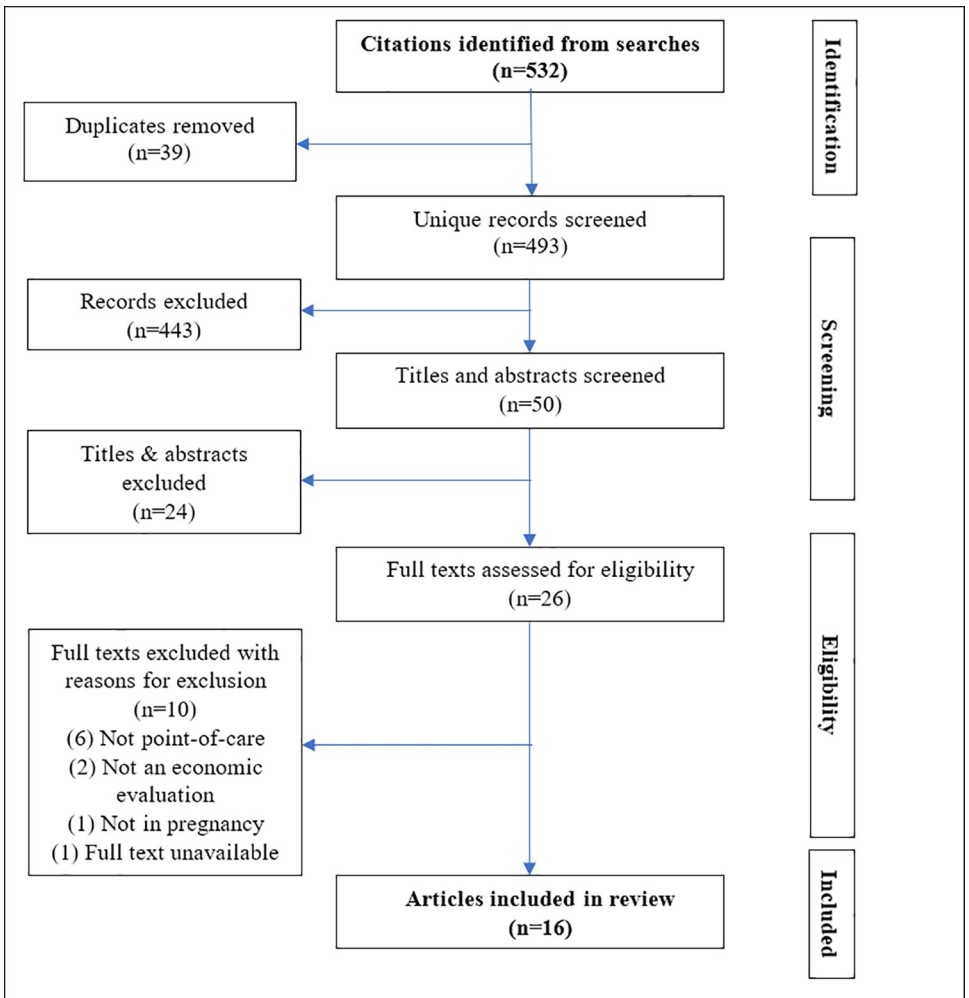

**Fig 1. PRISMA flow diagram of the study selection process for point-of-care testing and treatment for STIs in pregnancy in LMIC.**

related to tests and/or treatment, staff time, and other consumables such as medical supplies. Patient costs were rarely captured (n = 4) due to a provider perspective being taken by most studies.

All studies in this review compared a point-of-care testing and treatment intervention against one or more comparators as shown by Fig 2. Four studies compared laboratory-based testing with a point-of-care testing and treatment strategy [60, 63, 67, 70]. Six studies compared point-of-care testing and treatment against multiple comparators including laboratory testing, another point-of-care test, mass drug administration (MDA), syndromic management and/or no screening program [59, 61, 62, 68, 69, 73].

Most cost-effectiveness studies found point-of-care testing and treatment more cost-effective compared to laboratory-based testing [62, 67, 68, 70], no screening [58, 65, 66] or syndromic management [71] as shown in Table 4. For the full economic evaluations (n = 12), a variety of effects were measured. Most studies measured effectiveness by estimating the number of cases of adverse outcomes averted, such as miscarriage, stillbirth, neonatal death and congenital syphilis [58, 61, 65–71, 73].

**Table 3. Summary characteristics of economic evaluations for point-of-care tests for STIs in pregnancy in LMIC.**

| Author and Reference number | Study setting | Infection studied | Perspective | Comparators | Time Horizon | Cost components | Health outcomes | Efficiency measures |
|---|---|---|---|---|---|---|---|---|
| Partial Economic Evaluations | | | | | | | | |
| Shelley et al (2015) [64] | Zambia | Syphilis | Provider | Rapid Syphilis Test (RST) rollout vs. RST pilot | One year or less | 1. Test 2. Staff 3. Treatment 4. Supplies/ Consumables 5. Transport 6. Supervision 7. Quality Assurance and Control | Not applicable | Average cost per woman screened |
| Sweeney et al (2014) [63] | Tanzania | Syphilis | Provider | RST vs. Rapid Plasma Reagin (RPR) | One year or less | 1. Test 2. Staff 3. Treatment 4. Supplies/ Consumables 5. Training 6. Information Education Communication (IEC) material | Not applicable | Average cost per woman screened |
| Obure et al (2017) [72] | Colombia | Syphilis and HIV | Provider | Dual HIV and Syphilis point-of-care test (dRDT) vs. HIV point-of-care test (hRDT) and RST | One year or less | 1. Test 2. Staff 3. Treatment 4. Supplies/ Consumables | Not applicable | Average cost per woman screened |
| Levin et al (2007) [60] | Bolivia & Mozambique | Syphilis | Provider | RST vs. RPR | Not reported | 1. Test 2. Staff 3. Treatment 4. Supplies/ Consumables 5. Training 6. IEC material 7. Lab 8. Guidelines 9. Promotion | Not applicable | Average cost per woman screened |
| Full Economic Evaluations | | | | | | | | |
| Bristow et al (2016) [73] | Malawi | Syphilis | Societal | dRDT vs. 1. hRDT 2. hRDT and RST 3. hRDT and TPHA | Lifetime | 1. Test 2. Staff 3. Treatment 4. Patient out-of-pocket (OOP) expenses 5. Cost of delivery and immediate post-natal costs | Adverse pregnancy outcomes | Total cost & DALYs averted |
| Owusu-Edusei et al (2011) [61] | Sub-Saharan Africa | Syphilis | Societal and Provider | Dual RST (dRST) vs. 1. RST 2. Onsite RPR 3. Lab based RPR and TPHA 4. No screening | Lifetime | 1. Test 2. Staff 3. Treatment 4. Patient OOP expenses 5. Cost of delivery and immediate post-natal costs | Adverse pregnancy outcomes | Total cost & DALYs averted |

(*Continued*)

**Table 3.** (Continued)

| Author and Reference number | Study setting | Infection studied | Perspective | Comparators | Time Horizon | Cost components | Health outcomes | Efficiency measures |
|---|---|---|---|---|---|---|---|---|
| Kuznik et al (2015) [65] | Latin America and Asia | Syphilis | Provider | RST vs. No screening | Not reported | 1. Test 2. Staff 3. Treatment | 1. Neonatal death 2. Still birth 3. Congenital syphilis | incremental cost/DALY averted |
| Terris-prestholt et al (2015) [69] | Peru, Tanzania and Zambia | Syphilis | Provider | 1. RPR 2. RST 3. dRST (and treat all positives) 4. dRST (only treat if nTrp is positive) 5. RST followed by RPR 6. RPR followed by RST 7. RST followed by dRST (and treat all positives) 8. RST followed by dRST (only treat if nTrp is positive) 9. MDA vs. No screening | Not reported | 1. Test 2. Staff 3. Treatment 4. Supplies/ Consumables 5. Fixed clinic costs 6. RPR equipment 7. System costs | 1. Neonatal death 2. Still birth 3. Congenital syphilis | cost/DALY averted |
| Kuznik et al (2013) [58] | Sub-Saharan Africa | Syphilis | Provider | RST vs. No screening | Not reported | 1. Test 2. Staff 3. Treatment | 1. Neonatal death 2. Still birth 3. Congenital syphilis | average cost/ DALY averted |
| Rydzak and Goldie (2008) [62] | South Africa | Syphilis | Not reported | 1. RST 2. RPR confirmed with TPHA vs. No screening | Lifetime | 1. Test 2. Staff 3. Treatment 4. Supplies/ Consumables 5. Patient OOP expenses 6. Cost of delivery and immediate post-natal costs | 1. Neonatal death 2. Still birth 3. Congenital syphilis 4. Low birthweight (LBW) 5. Miscarriage | discounted costs saved per 1000 women |
| Schackman et al (2007) [68] | Haiti | Syphilis | Societal (CEA) and Provider (Scale up) | RST vs. 1. Syndromic management (in rural setting) 2. RPR (in urban setting) | Not reported | 1. Test 2. Staff 3. Treatment 4. Patient OOP expenses | 1. Neonatal death 2. Still birth 3. Congenital syphilis | Total incremental cost/DALY averted |
| Vickerman et al (2006) [70] | Tanzania | Syphilis | Not reported | RST (4 types of tests) vs. RPR | Not reported | 1. Test 2. Staff 3. Treatment | Adverse birth outcomes | total cost/DALY saved |
| Blandford et al (2007) [59] | South Africa | Syphilis | Provider | 1. Off-site RPR then TPHA; 2. Onsite RPR 3. RST vs. No screening | One year or less | 1. Test 2. Staff 3. Treatment 4. Supplies/ Consumables | Congenital syphilis | total incremental cost/cases averted |
| Mallma et al (2016) [67] | Peru | Syphilis | Not reported | RST vs. RPR | Not reported | 1. Test 2. Staff 3. Treatment 4. Supplies/ Consumables | Adverse birth outcomes | cost/DALY averted |

(*Continued*)

**Table 3.** (Continued)

| Author and Reference number | Study setting | Infection studied | Perspective | Comparators | Time Horizon | Cost components | Health outcomes | Efficiency measures |
|---|---|---|---|---|---|---|---|---|
| Romoren et al (2007) [74] | Botswana | Chlamydia | Provider | 1. syndromic management with Azithromycin treatment; 2. point-of-care testing with Erythromycin treatment; 3. point-of-care testing with Azithromycin treatment vs. Syndromic management with Erythromycin treatment | One year or less | Point-of-care testing and treatment: 1. Test 2. Staff 3. Treatment Syndromic management: 1. Staff 2. Treatment 3. Training 4. Supervision | 1. Neonatal death 2. Still birth 3. Congenital syphilis | incremental cost/cases cured |
| Larson et al (2014) [66] | Zambia | Syphilis | Provider | No screening program vs. 1. 62% of antenatal care attendees tested, only 10% of positive cases were treated; 2. 62% of antenatal care attendees tested, all positive cases were treated; and 3. All antenatal care attendees tested and all positive cases were treated | One year or less | 1. Test 2. Staff 3. Treatment 4. Supplies/ Consumables 5. Training | 1. Neonatal death 2. Still birth | total cost/DALY averted |

CEA: cost-effectiveness analysis; DALY: disability adjusted life year; dRDT: dual HIV and syphilis rapid diagnostic test; hRDT: HIV rapid diagnostic test; IEC: Information Education Communication; MDA: mass drug administration; nTrp: Non-treponemal; RST: rapid syphilis test; RPR: rapid plasma regain testing; TPHA: Treponema pallidum Hemagglutination Assay.

* Quality Control/ Quality Assurance refers to reviewing the quality of all the factors required for effective testing and treatment for syphilis in pregnancy.

** Larson et al (2014) utilised country representative statistics, from a previously conducted evaluation study, to build their scenarios. Country statistics showed that 62% of antenatal clinic attendees were tested for syphilis while only 10% of the test positives were treated.

Three-quarters of the full economic evaluations (n = 9) calculated incremental cost effectiveness ratios (ICERs), typically defined as the incremental cost per disability adjusted life years (DALY) averted or the incremental cost per case averted/cured [35, 37–40, 42, 43, 45, 46]. The ICERs reported varied considerably due to the heterogeneity of methods used to derive cost-effectiveness; therefore, making comparability difficult. The percentage differences emphasise the heterogeneity between studies included in this review. The greatest percentage difference between comparators was 170.6% (Cost/DALY of 63.1 and 5), while the smallest was 19.2% (Cost/DALY of 17 and 11.8).

Nine studies indicated a time horizon; the time horizon was typically restricted to 12 months or less [59, 63, 64, 66, 71, 72]. Only three studies modelled lifetime costs and effects of point-of-care testing and treatment for STIs in pregnancy on mothers and babies [61, 62, 73]. These studies did not explicitly indicate whether cost-effectiveness or outcomes were sustained over time.

## Key drivers of economic evaluations

A sensitivity analysis was performed in 14 studies (87.5%) by varying key assumptions and recording the impact this had on the findings of the evaluation. Ten studies conducted a

**Table 4. Summary results extracted from the economic evaluations for point-of-care tests for STIs in pregnancy in LMIC.**

| Author | Results | Cost-effectiveness decision rule | Drivers of cost and cost-effectiveness | Key findings | Generalisability of results | percentage difference |
|---|---|---|---|---|---|---|
| | | | Partial Economic Evaluations | | | |
| Shelley et al (2015) [64] | Pilot period<br>• Average unit cost/ woman tested: USD 1.49<br>• Average unit cost/ woman treated: USD 14.12<br>Rollout period<br>• Average unit cost/ woman tested: USD 4.84<br>• Average unit cost/ woman treated: USD 72.73 | Average cost per woman tested and treated is lower than the baseline average cost per woman tested and treated (Pilot program) | 1. Cost (RST test kit)<br>2. Screening coverage<br>3. Supply wastage | RST rollout had higher costs than RST pilot | Not stated | 105.5% (test); 135% (treated) |
| Sweeney et al (2014) [63] | RPR<br>• Average cost/woman tested: USD 2.32<br>• Average cost/woman treated: USD 12.96<br>RST<br>• Average cost/woman tested: USD 1.92<br>• Average cost/woman treated: USD 21.40 | Average cost per woman tested and treated is lower than the baseline average cost per woman tested and treated (RPR) | 1. Screening coverage<br>2. Supply wastage<br>3. Time taken to test | RST had higher costs than RPR | Not stated | 18.9% (test); 49.1% (treat) |
| Obure et al (2017) [72] | RST[a]<br>• Average Unit Cost/ woman tested: USD 10.26<br>• Average Unit Cost/ woman treated: USD 607.99.<br>dRDT<br>• Average Unit Cost/ woman tested: USD 15.89<br>• Average Unit Cost/ woman treated: USD 1859.26 | Average cost per woman tested and treated is lower than the baseline average cost per woman tested and treated (single hRDT and RST) | No sensitivity analysis | RST (and hRDT) had lower costs than dRDT | Not stated | 43.1% (test); 101.4% (treat) |
| Levin et al (2007) [60] | RPR<br>• Average Unit Cost/ woman tested: USD 1.43 (Bolivia) and USD 0.91 (Mozambique)<br>• Average Unit Cost/ woman treating: USD 40.09 (Bolivia) and USD 12.25 (Mozambique)<br>RST<br>• Average Unit Cost/ woman tested: USD 1.91 (Bolivia) and USD 1.05 (Mozambique)<br>• Average Unit Cost/ woman treated: USD 40.77 (Bolivia) and USD 13.45 (Mozambique) | Average cost per woman tested and treated is lower than the baseline average cost per woman tested and treated (RPR) | No sensitivity analysis | Mozambique: RST had higher costs than RPR<br>Bolivia: RST had lower costs than RPR | Not stated | 28.7% (for testing in Bolivia) and 1.7% (for treating in Bolivia);<br>14.3% (for testing in Mozambique) and 9.3% (for treating in Mozambique) |
| | | | Full Economic Evaluations | | | |

(*Continued*)

**Table 4.** (Continued)

| Author | Results | Cost-effectiveness decision rule | Drivers of cost and cost-effectiveness | Key findings | Generalisability of results | percentage difference |
|---|---|---|---|---|---|---|
| Bristow et al (2016) [73][b] | hRDT only<br>• Total cost: USD 21 875 298<br>• Effectiveness: 110 875 DALY<br>dRDT<br>• Total cost: USD 21 479 390<br>• Effectiveness: 108 693 DALY<br>hRDT and RST<br>• Total cost: USD 21 864 363<br>• Effectiveness: 110 691 DALY<br>hRDT and TPHA<br>• Total cost: USD 21 893 483<br>• Effectiveness: 110 697 DALY | Incremental costs and DALYs is lower than the baseline (RST & hRDT; HIV only; hRDT &TPHA) | 1. Prevalence<br>2. Screening coverage<br>3. Risk of adverse outcome | dRDT had lower costs and was more effective than hRDT; hRDT; and RST; hRDT and TPHA in lab | Results generalisable in similar countries | 1.9% (Total Cost); 2% (DALYs) |
| Owusu-Edusei et al (2011) [61][c] | Dual RST<br>• Total cost: USD 79 000<br>• Effectiveness: 5 adverse pregnancy outcomes<br>• DALYs: 42<br>RST<br>• Total cost: USD 76 000<br>• Effectiveness: 2 adverse pregnancy outcomes<br>• DALYs: 15<br>Onsite RPR<br>• Total cost: USD 84 000<br>• Effectiveness: 11 adverse pregnancy outcomes<br>• DALYs: 94<br>Offsite (lab based) RPR and TPHA<br>• Total cost: USD 86 000<br>• Effectiveness: 13 adverse pregnancy outcomes<br>• DALYs: 107<br>No screening<br>• Total cost: USD 106 000<br>• Effectiveness: 39 adverse pregnancy outcomes<br>• DALYs: 341 | Cost-savings are greater than the baseline and over-treatment rates are lower than the baseline | 1. Cost (RST test kit)<br>2. Test sensitivity (performance)<br>3. Cost associated with adverse pregnancy outcomes | dRST had lower costs but was less effective than RST; dRST had lower costs and was more effective onsite RPR; RPR and TPHA (lab); no screening | Not stated | 33% (Total Cost); 183.1% (DALYs) |
| Kuznik et al (2015) [65][d] | No screening program: Average ICER as not reported for either Asia or Latin America<br>RST:<br>Asia: average ICER was USD 53/ DALY averted; and<br>Latin America: average ICER was USD 60/ DALY averted | WHO threshold (ICER below country's Gross Domestic Product (GDP) per capita) | Prevalence | RST cost-effective compared to comparator | Not stated | No baseline figure provided to calculate percentage difference |

(*Continued*)

**Table 4.** (*Continued*)

| Author | Results | Cost-effectiveness decision rule | Drivers of cost and cost-effectiveness | Key findings | Generalisability of results | percentage difference |
|---|---|---|---|---|---|---|
| Terris-prestholt et al (2015) [69] | No screening program: ICER: Reported as not applicable Most cost-effective outcomes: 1. Peru: RST ICER was USD 53.69/ DALY averted; 2. Tanzania: MDA ICER was USD 8.7/ DALY averted; and 3. Zambia: MDA ICER was USD 5/ DALY averted | ICER is lower than the baseline scenario (no screening program) | 1. Cost (fixed clinic costs) 2. Screening coverage 3. RST reactivity rate 4. Antenatal clinic attendance | Peru: RST cost-effective compared to comparators; Tanzania: RST not as cost-effective as MDA; Zambia: RST not as cost-effective as MDA | Alludes to generalisability of results, but not definitively | 121.7% (for Peru); 167.5% (for Tanzania); 170.6 (for Zambia) |
| Kuznik et al (2013) [58][d] | No screening program: Average ICER was not reported RST Average ICER was USD 11/ DALY | Threshold of an ICER less than Gross National Income (GNI) per capita | Prevalence | RST cost-effective compared to comparator | Not stated | No baseline ICER provided to calculate percentage difference |
| Rydzak and Goldie (2008) [62] | No screening program • Discounted costs saved/ 1000 women: USD 110 220 RPR • Discounted costs saved/ 1000 women: USD 161 310 RST (dominates) • Discounted costs saved/ 1000 women: USD 170 030 | Cost-savings are greater than the baseline scenario (no screening program) | 1. Cost (labour, RST test kit price and other supplies) 2. Test sensitivity (performance) 3. Cost associated with adverse pregnancy outcomes | RST cost-effective compared to comparators | Results not generalisable to other settings or scenarios | 200% |
| Schackman et al (2007) [68] | Rural setting: RPR: ICER was USD 10.64/ DALY RST: ICER was USD 6.83/ DALY Urban setting: RST: ICER was USD9.95/ DALY | Thresholds: 1. ICER lower than USD50/ DALY recommended for health interventions in resource- poor countries; and 2. ICER lower than the GDP per capita of country where intervention is being implemented | 1. Prevalence 2. Test sensitivity (performance) | RST cost-effective compared to comparators | Results generalisable to HIV scale-up in resource poor settings | No baseline ICER provided to calculate percentage difference |
| Vickerman et al (2006) [70] | Test using serum: RPR • Cost/DALY: USD 12/ DALY RST (Determine) • Cost/DALY: USD 17/ DALY Test using whole blood: RPR • Cost/DALY: USD 16.8/ DALY RST (Determine) • Cost/DALY: USD 14.1/ DALY | Threshold cost-effectiveness ratio is lower than the baseline (RPR test) | Test sensitivity (performance) | RST cost-effective compared to comparators | Not stated | 19.2% (using serum) and 32.5% (using whole blood) |

(*Continued*)

**Table 4.** (Continued)

| Author | Results | Cost-effectiveness decision rule | Drivers of cost and cost-effectiveness | Key findings | Generalisability of results | percentage difference |
|---|---|---|---|---|---|---|
| Blandford et al (2007) [59] | Off-site RPR and confirmatory TPHA • Cases averted: 18 • Cost/case averted: USD 82 per case. Onsite RPR • Cases averted: 16 • Cost/case averted: not published (but states "dominated" RST • Cases averted: 27 • Cost/case averted: USD104 per case. | ICER is lower than the baseline scenario (no screening program) | 1. Prevalence 2. Test sensitivity (performance) 3. Relative distribution of active/latent syphilis | RST cost-effective compared to comparators | Not stated | 33.8% |
| Mallma et al (2016) [67] | RST • Cost/DALY averted: USD 46/ DALY averted RPR • Cost/DALY averted: USD 109/ DALY averted | WHO threshold (ICER is below the country's GDP per capita) | 1. Prevalence 2. Cost (salaries) 3. Test positivity rates 4. Screening coverage 5. Number of hours worked | RST cost-effective compared to comparator | Not stated | 81.3% |
| Romoren et al (2007) [74] | Syndromic Management • Cases cured (erythromycin): 800 • Cost/ cured case (erythromycin): USD 66 • Cases cured (azithromycin): 1500 • Cost/ cured case (azithromycin): USD 21 point-of-care test (0.75 sensitivity- base case) • Cases cured (erythromycin): 3200 • Cost/ cured case (erythromycin): USD 35 • Cases cured (azithromycin): 3500 • Cost/ cured case (azithromycin): USD 31 | Willingness-to-pay threshold (cost-effectiveness ratio is lower than the willingness to pay threshold) | 1. Prevalence 2. Cost (point-of-care test) 3. Test sensitivity (performance) 4. Probability of partner notification | point-of-care test combined with Azithromycin treatment cost-effective compared to comparators | Results generalisable to other sub-Saharan countries | 62.5% (using erythromycin) and 94.9% (using azithromycin) |
| Larson et al (2014) [66] | Evaluation study conditions (62% of antenatal clinic (ANC) attendees tested and 10% of positive cases treated): • USD628/ DALY averted 62% of ANC attendees tested, and all positive cases treated: • USD66/ DALY averted All ANC attendees tested, and all positive cases treated: • USD60/ DALY averted | ICER is lower than the baseline scenario | 1. Prevalence 2. Cost (training) 3. Number of antenatal clinic attendees tested and treated 4. Staff time | RST (testing and treating all antenatal clinic attendees) cost-effective compared to comparators) | Results not generalisable to other settings or scenarios | 165.1% |

ANC: Antenatal clinic; CEA: cost-effectiveness analysis; DALY: Disability-adjusted-life-years; GDP: Gross domestic product; GNI: Gross national income; dRDT: dual HIV and syphilis rapid diagnostic test; hRDT: HIV rapid diagnostic test; ICER: Incremental cost effectiveness ratio; IEC: Information Education Communication; MDA: mass drug administration; nTrp: Non-treponemal; RST: rapid syphilis test; RPR: rapid plasma regain testing; WHO: World Health Organization.

a. Only syphilis was included in the analysis, results for HIV were excluded.

b. Did not publish an ICER (or Cost/DALY) as the difference between each comparator would be smaller than 1, and therefore not significant.

c. Did not publish an ICER (or Cost/DALY), as it may not have yielded a meaningful outcome.

d. Reported average ICERS.

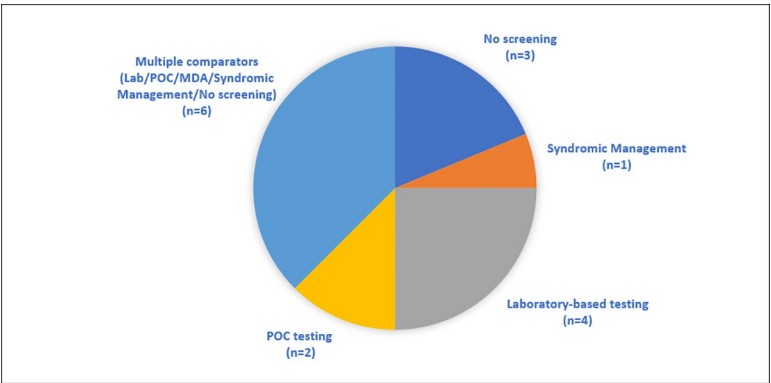

**Fig 2. Economic evaluation comparators for point-of-care testing and treatment for STIs in pregnancy in LMIC.**
MDA: Mass Drug Administration; POC: point-of-care.

univariate sensitivity analysis and of those studies, six conducted a multivariate sensitivity analysis [59, 61, 63, 64, 68, 73] and three conducted a probabilistic sensitivity analysis [58, 69, 71] in addition to the univariate sensitivity analysis. The main drivers for costs and cost effectiveness are illustrated in Table 4 and included: STI prevalence [58, 59, 65–69, 71, 73], cost of test, cost of treatment, cost of training or salaries and wages [58, 61, 62, 64, 66–69, 71], test sensitivity and specificity [58, 59, 61, 62, 68, 70, 71], and treatment and/or screening coverage [63, 64, 66, 69, 73].

## Quality appraisal

All studies met 75% or more of the methodology criteria of the Drummond Checklist and 60% for the CHEERS checklist (Fig 3). Fig 3 illustrates the shortcomings with respect to reporting criteria in the reviewed papers. The appraisal based on the Drummond checklist (Table 5), indicated that a key gap was the reporting of all relevant costs with few studies taking account of training, quality control and quality assurance. The appraisal using the CHEERS checklist (Table 6) highlighted that only half the studies (n = 8) clearly outlined the time horizon for the

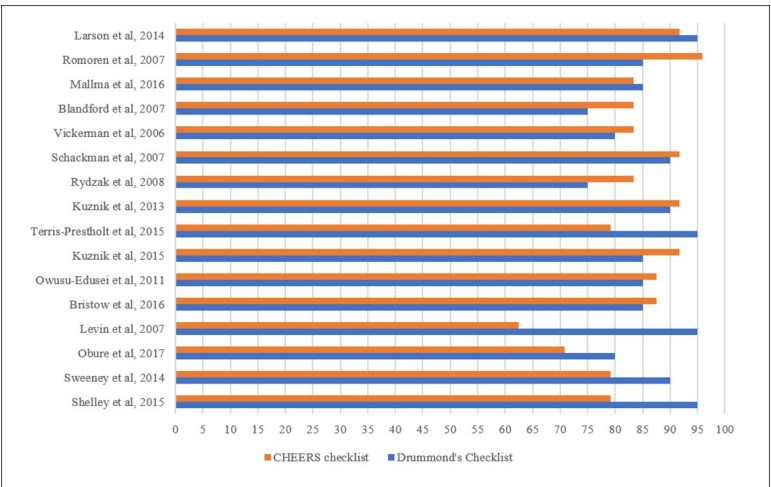

**Fig 3. Assessment of methodological and reporting quality of economic evaluations of point-of-care testing and treatment for STIs in pregnancy in LMIC (%).**

**Table 5. Drummond 10-point checklist.**

| | | References | | | | | | | | | | | | | | | |
|---|---|---|---|---|---|---|---|---|---|---|---|---|---|---|---|---|---|
| | | [65] | [69] | [58] | [62] | [68] | [70] | [59] | [67] | [71] | [66] | [64] | [63] | [72] | [60] | [73] | [61] |
| 1 | Well defined research question stated | 1 | 1 | 1 | 1 | 1 | 1 | 1 | 0.5 | 1 | 1 | 0.5 | 1 | 1 | 1 | 1 | 1 |
| 2 | Comprehensive description of competing alternatives | 1 | 1 | 1 | 1 | 1 | 1 | 1 | 1 | 1 | 1 | 1 | 1 | 1 | 1 | 1 | 1 |
| 3 | Evidence of program effectiveness included | 1 | 1 | 1 | 1 | 1 | 1 | 1 | 1 | 1 | 1 | 1 | 1 | 1 | 1 | 1 | 1 |
| 4 | All relevant cost and consequences for each alternative identified | 0.5 | 1 | 0.5 | 0 | 0.5 | 0 | 0.5 | 0 | 0.5 | 1 | 1 | 1 | 0.5 | 1 | 0.5 | 0.5 |
| 5 | Costs and consequences measured accurately and appropriately | 0.5 | 1 | 1 | 1 | 1 | 1 | 1 | 1 | 1 | 1 | 1 | 1 | 1 | 1 | 1 | 1 |
| 6 | Costs and consequences valued credibly | 1 | 1 | 1 | 0.5 | 1 | 1 | 0.5 | 1 | 1 | 1 | 1 | 1 | 1 | 1 | 1 | 1 |
| 7 | Costs and consequences adjusted for differential timing | 1 | 1 | 1 | 1 | 1 | 0 | 0 | 1 | 0 | 1 | 1 | 0.5 | 0 | 1 | 0.5 | 0.5 |
| 8 | Incremental analysis of costs and consequences performed | 1 | 1 | 1 | 1 | 1 | 1 | 1 | 1 | 1 | 1 | 1 | 1 | 1 | 1 | 1 | 1 |
| 9 | Allowance made for uncertainty in cost and consequence estimates | 0.5 | 0.5 | 0.5 | 0.5 | 0.5 | 1 | 0.5 | 1 | 1 | 0.5 | 1 | 0.5 | 0.5 | 0.5 | 0.5 | 0.5 |
| 10 | Presentation/Discussion included all concerns raised in the results | 1 | 1 | 1 | 0.5 | 1 | 1 | 1 | 1 | 1 | 1 | 1 | 1 | 1 | 1 | 1 | 1 |
| | Score | 8.5 | 9.5 | 9 | 7.5 | 9 | 8 | 7.5 | 8.5 | 8.5 | 9.5 | 9.5 | 9 | 8 | 9.5 | 8.5 | 8.5 |

In this Table 1 denotes that the checklist item is clearly included in the study; 0, that the checklist item is not included in the study; and 0.5, that although the item is present, it is not clear.

analysis [59, 61–64, 66, 71, 72], most taking a one year or shorter time horizon [59, 63, 64, 66, 71, 72]. Half the studies (n = 8) utilised analytical methods to deal with missing values and uncertainty [58, 61, 63, 65, 67, 68, 70, 71]. Only a third of the studies (n = 6) explained variations in the data by different population characteristics [60, 64, 67, 70–72]. The most common sub-group analyses were by type or location of health facility (hospital or health centre and urban or rural).

## Discussion

This review identified 16 economic evaluations set in LMIC on point-of-care testing and treatment for STIs in pregnancy and presented a synthesis of the evidence. All but one study focused on syphilis and most were set in African countries. The majority of studies in this review suggest that point-of-care testing and treatment for syphilis and CT in pregnancy can be cost-effective in LMIC settings when compared to no screening programs, laboratory-based testing and/or syndromic management. These studies also indicate that point-of-care testing and treatment for STIs is most cost-effective where access to alternative testing mechanisms is limited, including laboratory testing facilities. Further, there was considerable variation in the types of costs and outcomes utilised by economic evaluations, as well as in the time horizons and sample sizes, which proved to be a challenge in comparing the costs and cost effectiveness of interventions across the different settings. The decision rules for cost-effectiveness and percentage differences in Table 4 demonstrate the between study variation. The decision rules indicate that cost-effectiveness is dependent upon the decision rule utilised. Further, percentage difference highlights the difference between the largest and smallest cost-effectiveness measures per study. The largest percentage difference between comparators was 170% [69] while the smallest 19% [70], which demonstrates the large variability with respect to measures of cost-effectiveness. Key drivers of cost and cost-effectiveness were STI prevalence and costs (including the cost of tests, treatment, training costs, and salaries or wages.

Our review also highlighted some gaps in the evidence, in terms of methodology and scope. First, the evidence is limited both in geographic and infection scope. Most studies were

**Table 6. Consolidated Health Economic Evaluation Reporting Standards (CHEERS) checklist.**

| Section/Item (including item number) on CHEERS checklist | | | [65] | [69] | [58] | [62] | [68] | [70] | [59] | [67] | [71] | [66] | [64] | [63] | [72] | [60] | [73] | [61] |
|---|---|---|---|---|---|---|---|---|---|---|---|---|---|---|---|---|---|---|
| | | | | | | | | | | | References | | | | | | | |
| Title and abstract | 1 | Title | Y | Y | Y | Y | Y | Y | Y | Y | Y | Y | Y | Y | Y | Y | Y | Y |
| | 2 | Abstract | Y | Y | Y | Y | Y | Y | Y | Y | Y | Y | Y | Y | Y | Y | Y | Y |
| Introduction | 3 | Background | Y | Y | Y | Y | Y | Y | Y | Y | Y | Y | Y | Y | Y | Y | Y | Y |
| Methods | 4 | Target pop. & sub-groups | Y | Y | Y | Y | Y | Y | Y | Y | Y | Y | Y | Y | Y | Y | Y | Y |
| | 5 | Setting and location | Y | Y | Y | Y | Y | Y | Y | Y | Y | Y | Y | Y | Y | Y | Y | Y |
| | 6 | Study perspective | Y | Y | Y | Y | Y | N | Y | N | Y | Y | Y | Y | Y | Y | Y | Y |
| | 7 | Comparators | Y | Y | Y | Y | Y | Y | Y | Y | Y | Y | Y | Y | Y | Y | Y | Y |
| | 8 | Time horizon | Y | N | N | Y | N | N | Y | N | Y | Y | Y | Y | Y | N | N | Y |
| | 9 | Discount rate | Y | Y | Y | Y | Y | N | N | Y | N | Y | Y | Y | N | Y | Y | Y |
| | 10 | Health outcomes | Y | Y | Y | Y | Y | Y | Y | Y | Y | Y | N | N | N | N | Y | Y |
| | 11a | Effectiveness measures: single-study estimates | NA | Y | NA | NA | NA | Y | NA | N | NA | NA | NA | NA | NA | NA | NA | NA |
| | 11b | Effectiveness measures: synthesis-based estimates | Y | NA | Y | Y | Y | NA | Y | N | Y | Y | NA | NA | NA | NA | Y | Y |
| | 12 | preference based outcome measurement/valuation | Y | Y | Y | Y | Y | Y | Y | Y | Y | Y | NA | NA | NA | NA | Y | Y |
| | 13a | Resource/Cost estimates: single study-based | Y | NA | NA | NA | NA | NA | Y | Y | NA | Y | Y | Y | Y | Y | NA | NA |
| | 13b | Resource/Cost estimates: model-based | NA | Y | Y | Y | Y | Y | NA | NA | Y | NA | NA | NA | NA | NA | Y | Y |
| | 14 | Currency, price data and conversion | Y | Y | Y | Y | Y | Y | Y | Y | Y | Y | Y | Y | Y | Y | Y | Y |
| | 15 | Model choice | Y | N | Y | Y | Y | N | Y | N | Y | Y | Y | Y | Y | Y | Y | Y |
| | 16 | Assumptions | N | Y | Y | Y | Y | Y | Y | Y | Y | Y | Y | Y | Y | N | Y | Y |
| | 17 | Analytical methods | Y | N | Y | N | Y | Y | N | Y | Y | N | N | Y | N | N | N | Y |
| Results | 18 | Study parameters | Y | Y | Y | Y | Y | Y | Y | Y | Y | Y | Y | Y | Y | Y | Y | Y |
| | 19 | Incremental costs and outcomes | Y | Y | Y | Y | Y | Y | Y | Y | Y | Y | NA | NA | NA | NA | Y | Y |
| | 20a | Characterising uncertainty: single study-based | Y | Y | Y | Y | Y | Y | Y | Y | NA | Y | Y | Y | N | N | Y | Y |
| | 20b | Characterising uncertainty: model-based | NA | NA | NA | NA | NA | NA | NA | NA | Y | NA | NA | NA | N | N | NA | NA |
| | 21 | Characterising heterogeneity | N | N | N | N | N | Y | N | Y | Y | N | Y | N | Y | Y | N | N |
| Discussion | 22 | Summary key findings | Y | Y | Y | Y | Y | Y | Y | Y | Y | Y | Y | Y | Y | Y | Y | Y |
| Other | 23 | Funding source | Y | Y | Y | N | Y | Y | Y | Y | Y | Y | Y | Y | Y | Y | Y | N |
| | 24 | Conflicts of interest | Y | Y | Y | N | Y | Y | N | Y | Y | Y | Y | Y | Y | N | Y | N |

Y: Yes (included in the article); N: No (Not included in the article); NA: Not applicable (not applicable to the type of study).

conducted on the African continent, with only one study conducted in Asia [65] and none in the Pacific. Further, point-of-care testing and treatment for syphilis in pregnancy dominated the literature. Only one economic evaluation focussed on CT and there were no economic evaluations of other prevalent, curable STIs such as NG, TV or BV. The lack of studies from Asia-Pacific countries is of concern given the high STI prevalence, particularly for common infections including syphilis, NG, TV, CT, reported among pregnant women in the region [68, 72, 75] as well as the widespread integration of point-of-care testing and treatment for syphilis in antenatal clinics [22]. Although existing economic evaluations provide guidance to cost-effectiveness, context matters. In particular, results from economic evaluations cannot always be generalised because local disease epidemiology, social, cultural and financial barriers to implementation vary between settings [76]. These could lead to differences in the uptake of testing and treatment between settings, and therefore, also differences in the cost of scale up.

In the last decade, the number of studies based in LMICs evaluating the utility of point-of-care tests for the detection and treatment of STIs in pregnancy has increased [38, 77], which has influenced the number of economic evaluations conducted on these types of tests. Yet with few cheap, accurate and easy to use point-of-care tests for CT, NG, TV and BV available [78], there have not been many effectiveness trials conducted in LMIC. As such, the data available to conduct robust economic evaluations is limited. Therefore, there is a need to expand the scope of effectiveness studies and accompanying economic evaluations of point-of-care testing for high-burden STIs in pregnancy [38, 79, 80].

Second, most studies included in this review took the provider perspective. Only three studies took a societal perspective. The paucity of studies conducted from the societal perspective potentially excludes and/or underestimates the direct and indirect costs to women such as reducing patient waiting times and reducing the number of clinic visits required to receive results, which in turn is expected to reduce the considerable direct and indirect costs incurred by many women seeking antenatal care in LMICs [81, 82]. Consequently, these direct and indirect costs can be, in many settings, barriers to participation as costs are often borne to the patient as OOP expenses and can lead to catastrophic health expenditure, which might lead to a drop in uptake/participation.

Third, most economic evaluations reviewed had short time horizons of 12 months or less. Short time horizons do not consider the lifetime implications of effective point-of-care testing and treatment of STIs in pregnancy. Economic evaluations in this field should ideally incorporate the long-term costs and benefits to both mother and baby from adopting point-of-care testing in order to capture said costs and benefits that occur later in life.

Our review indicated that all but two studies conducted a sensitivity analysis, which demonstrate the variables that drive costs and cost-effectiveness in each study. The most common drivers of cost and cost-effectiveness were STI prevalence and costs (which included the cost of tests, treatment, training costs, and salaries or wages). The sensitivity analyses of the studies included in this review did not explore the potential variation in loss-to-follow-up. These findings are consistent with other systematic reviews that have indicated that HIV prevalence and the cost of the HIV test are key drivers of cost-effectiveness of HIV screening in pregnancy [83] and key populations, such as sex workers [84].

We also found that most studies applied a narrow definition to the costs of point-of-care testing including only the cost of test kits, treatment costs and staff salaries. Costs associated with other core activities such as quality control, quality assurance and procurement are not negligible, and thus these studies may have underestimated the true cost of POC testing [85]. Underestimating costs leads to inaccurate projected costs for implementation and scale up, which consequently means that insufficient budget will be allocated to the implementation and scale up of the intervention. Insufficient funding, will inevitably mean that scale up will not go as planned, may be suspended, or may not run successfully [45]. Standardising the methodology, or introducing guidelines, for measuring costs may mitigate the likelihood that crucial costs are excluded from analyses. In addition, this will aid the comparison of costs across studies, which was particularly difficult in this review.

Finally, increased attention should also be paid to the broader issues of affordability and equity. Despite recent recommendations by the International Society for Pharmacoeconomics and Outcomes Research to conduct budget impact analyses alongside economic evaluations [45], no studies in this review did so although two did mention its importance [58, 62]. Budget impact analyses explore economic and financial consequences highlighting the affordability of program implementation. They specifically factor in short-run costs to cater for government budget allocations [45, 86] and are especially relevant in settings where health care resources are highly constrained [49]. There have been ongoing calls for incorporating equity within

economic evaluations of health interventions [46, 48]. This is particularly relevant for point-of-care testing and treatment, which can reduce inequalities in healthcare by diagnosing and treating STIs in a single visit to a health facility. This is important in populations with limited access to laboratory facilities [87]. Future studies in this field should explore ways to expand cost-effectiveness analysis to address health equity concerns including distributional cost-effectiveness analysis which quantifies the distribution of costs and effects by equity-relevant variables such as socioeconomic status, location, ethnicity, sex and severity of illness [88]. Alternatively, non-health benefits such as financial risk protection could be measured using extended cost-effectiveness analysis [47, 89].

Economic evaluations generate evidence to optimally allocate resources and provide the foundation for the delivery of healthcare interventions [55]. This review has highlighted that point-of-care testing for STIs in pregnancy has proven to be cost effective in African settings. The results may be generalisable to similar settings, however consideration should be taken with the shortcomings revealed by this review, including the heterogenous nature of economic evaluations, the perspective taken and the short run time horizons. Suggesting that evidence should be interpreted with caution, and an economic evaluation conducted prior to implementing programmes to fill these gaps and provide better information to aid uptake, scale up and indicate priority settings in resource constrained contexts. Going forward, extended cost-effectiveness analyses and budget impact analyses provide evidence for the implementation of a cost-effective intervention, which is equitable, affordable, and sustainable. Extended cost-effectiveness analyses highlight the distributional effects within a given populations as a result of the implementation of an intervention; while budget impact analyses determine the investment required by the government to implement an intervention; and demonstrate affordability given available resources and budget constraints.

This review has some limitations, with respect to the synthesis of key results, the significant variability in economic methods, particularly regarding the types of costs and outcomes measured, meant that the planned meta-analysis was not possible [54]. Further, few studies utilised a societal perspective to determine costs, which may have understated the true value associated with point-of-care testing and treatment for STIs in pregnancy. However, the descriptive and narrative synthesis provides valuable insight into these limitations and gaps and suggests direction for future research. Additionally, most studies took a narrow view of costs and measured them along a short-run time horizon, which highlights a short-coming of the studies included in this review. This systematic review did not include grey literature, which can also be an important source of evidence but less likely to be subject to independent scientific peer-review, where additional bias is introduced. Finally, a further enhancement of economic evaluations would be the inclusion of equity and affordability analyses.

## Conclusion

Our review indicates that point-of-care testing and treatment for syphilis in pregnancy is cost-effective in LMICs compared to laboratory-based testing, syndromic management and where no testing programs have been implemented. The review also revealed that key drivers of cost and cost-effectiveness are STI prevalence and costs (including the cost of tests, treatment, training costs, and salaries or wages. It also highlights important gaps in the published literature that requires urgent attention by researchers. These gaps include broadening the scope of economic evaluations to include common curable STIs, such as CT, NG, TV and BV; the geographical representation of studies; the costs borne to pregnant women and their families as a result of accessing testing and treatment for STIs; the life time costs and effects of testing and treatment on mothers and their babies and finally; the budget and equity

implications of implementing point-of-care testing and treatment versus other screening practices.

## Supporting information

**S1 File. Preferred Reporting Items for Systematic Reviews and Meta-Analyses (PRISMA) checklist.**
(DOCX)

**S1 Dataset.**
(XLSX)

## Author Contributions

**Conceptualization:** Olga P. M. Saweri, Neha Batura, Rabiah Al Adawiyah, Louise M. Causer, William S. Pomat, Andrew J. Vallely, Virginia Wiseman.

**Data curation:** Olga P. M. Saweri, Neha Batura, Louise M. Causer.

**Formal analysis:** Olga P. M. Saweri, Neha Batura, Louise M. Causer, Virginia Wiseman.

**Funding acquisition:** William S. Pomat, Andrew J. Vallely.

**Investigation:** Olga P. M. Saweri, Neha Batura, Rabiah Al Adawiyah, William S. Pomat, Andrew J. Vallely, Virginia Wiseman.

**Methodology:** Olga P. M. Saweri, Neha Batura, Louise M. Causer, Virginia Wiseman.

**Project administration:** Olga P. M. Saweri, Neha Batura, Andrew J. Vallely, Virginia Wiseman.

**Resources:** William S. Pomat, Andrew J. Vallely, Virginia Wiseman.

**Software:** Olga P. M. Saweri, Neha Batura.

**Supervision:** Olga P. M. Saweri, Andrew J. Vallely, Virginia Wiseman.

**Validation:** Andrew J. Vallely, Virginia Wiseman.

**Visualization:** Olga P. M. Saweri, Neha Batura, Rabiah Al Adawiyah.

**Writing – original draft:** Olga P. M. Saweri, Neha Batura, Louise M. Causer, Virginia Wiseman.

**Writing – review & editing:** Olga P. M. Saweri, Neha Batura, Rabiah Al Adawiyah, Louise M. Causer, William S. Pomat, Andrew J. Vallely, Virginia Wiseman.

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
