## [Decision Letter · Decision Letter 0]

3 Nov 2020

PONE-D-20-31634

Economic evaluation of point-of-care testing and treatment for sexually transmitted and genital infections (STIs) in pregnancy in low- and middle-income countries (LMICs): A systematic review

PLOS ONE

Dear Dr. Saweri,

Thank you for submitting your manuscript to PLOS ONE. After careful consideration, we feel that it has merit but does not fully meet PLOS ONE’s publication criteria as it currently stands. Therefore, we invite you to submit a revised version of the manuscript that addresses the points raised during the review process.

This is a well-written manuscript. I concur with the reviewers' comments.

We look forward to receiving your revised manuscript.

Kind regards,

Remco PH Peters, MD, PhD, DLSHTM

Academic Editor

PLOS ONE

Journal Requirements:

2. Please amend either the title on the online submission form (via Edit Submission) or the title in the manuscript so that they are identical.

Reviewers' comments:

Reviewer's Responses to Questions

**Comments to the Author**

1. Is the manuscript technically sound, and do the data support the conclusions?

Reviewer #1: Yes

Reviewer #2: Yes

2. Has the statistical analysis been performed appropriately and rigorously? 

Reviewer #1: N/A

Reviewer #2: Yes

3. Have the authors made all data underlying the findings in their manuscript fully available?

Reviewer #1: Yes

Reviewer #2: Yes

4. Is the manuscript presented in an intelligible fashion and written in standard English?

Reviewer #1: Yes

Reviewer #2: Yes

5. Review Comments to the Author

Reviewer #1: GENERAL/MAJOR COMMENTS

This systematic review of economic evaluations of point of care STI tests is well-designed, well-conducted and generally well-described.

However, the critical interpretation might be deepened – both in (A) the descriptive analysis in Table 3 and (B) the Discussion.

(A) Failing the option to do a formal statistical meta-analysis (due to heterogeneity in outcome measures and …), the crux of the findings are in the descriptive Table 3. In this table, however, between Results and outcome/conclusion the relation is not always clear, nor are the comparative outcome definitions under Results always clear. Please see Specific comments below.

(B) The discussion dwells on methods of economic evaluations and to what extend the existing, reviewed studies met or not the optimum standards. Apart from this partly academic matter, what is the more practical conclusion from the presented evidence? Concretely, if a health minister (in country X) considers changing its policy from syndromic treatment to point-of-care testing (for, say, syphilis), what could s/he learn from this review? Should a new (local?) study be carried out, and if so what would be the preferred methodology and design? Or is the evidence in favor of POC testing already clear and uncontested, in all or some (which=?) settings?

SPECIFIC COMMENTS

RESULTS: Table 3:

Shelley et al: What is meant with ‘Quality control measures valuable to scale-up of RST’? How does the ‘quality control’ relate to the Results? Please expand the Results such that this becomes clear. Also – if space allows – comment on how rural and urban settings differed, e.g. which setting had higher syphilis prevalence, or what was the driver of the comparative cost and screening and treatment volumes?

Bristow et al: What was the criterion for ‘best option’? I suppose, cost per DALY averted? If so, please add ‘cost per DALY averted’ to the results (possibly replacing cost and DALY). Or if you keep the outcome DALY: is this DALYs lost or DALYs averted?

Adding an outcome per infection or DALY averted should help to make the study summary more comparable to other studies higher and lower within Table 3.

Owusu-Edusei et al: Same comment as for Bristow, on cost versus cost per DALY, and the definition if ‘more effective’, and of DALY (lost or averted).

Terris-Prestholt et al.: The ‘No screening program ; ICER reported as not applicable’ is redundant, as this is the definition of the counterfactual/comparator, ICERs are expressed against this counterfactual/comparator.

Kuznik et al: ICER is for what? Per infection averted? Please write that out. And clarify, what does an ICER mean for the ‘No screening program’, what was the comparator or counterfactual scenario here?

• Rydzak: To interpret the ‘RST dominates’, please write out what is the comparator scenario for each scenario (No screening; RPR, and RST), for example:

No screening compared to …

• RPR compared to No screening

• RST compared to ?RPR? or ?No screening?

Schackman et al. Please clarify how Rural and Urban differed, e.g. in prevalence? Without that info, the differential results are meaningless to the reader.

Vickerman et al: From the 4 outcomes (cost/DALY) given, RPR has lower cost/DALY for Serum but RST has lower cost/DALY for Whole Blood. Therefore the conclusion ‘test using serum/whole blood: Cost-effective outcome = RST’ is not obvious from the results – the results suggest RST as more cost-effective for whole blood but not for serum. Please clarify; for example, add into the Results cell, the corresponding cost per DALY for Serum + Whole Blood combined.

Blandford: As for Kuznik et al., above, please clarify the respective comparators for each of the 3 testing options. Also clarify ‘Not published but states ‘Dominiated’.

To facilitates readers’ matching between Conclusion and Results, I’d rephrase the Conclusion as ‘Off-site RPR and confirmatory TPHA’. And remove ‘(averts more cases)’ as that’s obvious from, and just a repeat of, the Results.

Romoren et al: From the results, between the 4 options, the most cost effective looks to be Syndromic Management with azithromycin, at USD 21. In spite of that result, the conclusion states POC with azithromycin, despite it costing USD31 per cured case. Please reconciliate and clarify.

Larson et al.: Where does the 62% come from, what algorithm or policy was this the result of?

At reading the conclusion, I infer back that ‘RST’ must correspond to test and treat all, i.e. the third of the 3 options. Please write this out in the Results. And perhaps shorten the corresponding scenario names in column Intervention versus Comparator.

DISCUSSION

The Discussion mentions many wise and true things, but they are not too well structured, leaving the reader without a clear understanding of the overall current evidence or its policy implications.

We would recommend structuring the discussion of limitations between:

(1) Limitations that affect the core outcomes of studies analyzed (in Table 3), i..e. that bias those comparisons, notably:

• Societal perspective, which may cause under-stating the value POC compared to laboratory-based policies, if the savings from POC are for a large part on the patient/client side.

• Types of cost: if the variability across studies is important, then please show it – adding a column with ‘Cost components included’ to Table 3. (To make space in Table 3, you might remove the column Intervention versus Comparator’ which gets repeated in the next-right column in any case.?)

(2) More general limitations or qualifications such as:

• Most studies costed only test kits, treatment and staff, but not QAQC or procurement: if this applies across all scenarios compared within a study, that comparison is not biased.

• Affordability and equity: True but somewhat beyond the scope of this review?

• Short time horizons: A true and important limitation – but not one that biases the comparison across scenarios within a study: Per treatment or maternal infections cured, the longer-term effects should be similar regardless of through which screening algorithm that treatment or cure was produced, not?

CONCLUSION:

The start and end of the conclusion as written are more an Introduction. I suggest to narrow and focus the conclusion on the part in the middle, from ‘Our review indicates…’ to ‘… other screening practices’.

Regarding the lack of geographical representation of the studies:

Why is that a problem? Is there reason to think that the alternative screening algorithms would compare differently in different settings (e.g. in Latin America or Asia-Pacific)? Why, what setting-specific determinant (e.g. STI prevalence??) would be the driver of varying cost-effectiveness rankings? If you believe this is truly important, then make it more concrete, beyond a description of geographic representation, in the 3rd paragraph of the Discussion.

Reviewer #2: Please see attached comments.

This manuscript provides the results of a systematic review of economic evaluations of antenatal point-of-care (PoC) testing and treatment for sexually transmitted infections (STIs) in low- and middle-income countries (LMICs). The authors identified 16 economic evaluations in the peer reviewed literature that met the inclusion criteria. Because of heterogeneity, authors were only able to provide a summary of the literature, and concluded that PoC testing and treatment during pregnancy is likely to be cost-effective compared to no testing, syndromic management, and lab-based testing. This is an important topic likely to be of interest to the readers of PLOS ONE. Most countries continue to provide sub-optimal antenatal STI care through the syndromic approach, missing infections and overtreating pregnant women. A major barrier to expanding etiological testing is cost and the uncertainty around the cost-effectiveness of testing. I suggest accepting this manuscript with some corrections and clarifications outlined below.

6. PLOS authors have the option to publish the peer review history of their article (what does this mean?). If published, this will include your full peer review and any attached files.

Reviewer #1: No

Reviewer #2: **Yes: **Adriane Wynn

---

## [Author Response · Author response to Decision Letter 0]

22 Jan 2021

Responses to Reviewers 

Reviewer #1 comments:

Failing the option to do a formal statistical meta-analysis (due to heterogeneity in outcome measures and …), the crux of the findings are in the descriptive Table 3. In this table, however, between Results and outcome/conclusion the relation is not always clear, nor are the comparative outcome definitions under Results always clear. Please see Specific comments below.

RESPONSE: Thank you for your comment. The results and outcomes presented in Table 3 are results published by each study included in the review. Table 3 in the manuscript has been revised to clarify the relationship between the results and outcome/conclusion. The table now have 5 columns that present cost components, health outcomes (where applicable), efficiency measures, the drivers of cost effectiveness, and key findings of each economic evaluation. Each column has been systematically completed to improve coherence and comparability across the studies included in this review (see pages 9-17, line #190).

The discussion dwells on methods of economic evaluations and to what extend the existing, reviewed studies met or not the optimum standards. Apart from this partly academic matter, what is the more practical conclusion from the presented evidence? Concretely, if a health minister (in country X) considers changing its policy from syndromic treatment to point-of-care testing (for, say, syphilis), what could s/he learn from this review? Should a new (local?) study be carried out, and if so what would be the preferred methodology and design? Or is the evidence in favor of POC testing already clear and uncontested, in all or some (which=?) settings?

RESPONSE: Thank you for your comment, we have edited the discussion to explain a practical recommendation for implementation (see page 27-28, lines #354-369).

Specific Comments:

RESULTS: Table 3:

1) Shelley et al: What is meant with ‘Quality control measures valuable to scale-up of RST’? How does the ‘quality control’ relate to the Results? Please expand the Results such that this becomes clear. Also– if space allows – comment on how rural and urban settings differed, e.g. which setting had higher syphilis prevalence, or what was the driver of the comparative cost and screening and treatment volumes?

RESPONSE: The term ‘quality control’ is listed in column 8 of Table 3 and is now explained in the footnotes of Table 3 (denoted by *). 

2) Bristow et al: What was the criterion for ‘best option’? I suppose, cost per DALY averted? If so, please add ‘cost per DALY averted’ to the results (possibly replacing cost and DALY). Or if you keep the outcome DALY: is this DALYs lost or DALYs averted?

Adding an outcome per infection or DALY averted should help to make the study summary more comparable to other studies higher and lower within Table 3. 

RESPONSE: This has now been made clearer through the inclusion of columns 8 and 9 of Table 3.

3) Owusu-Edusei et al: Same comment as for Bristow, on cost versus cost per DALY, and the definition if ‘more effective’, and of DALY (lost or averted). 

RESPONSE: This has now been made clearer through the inclusion of columns 8 and 9 of Table 3.

4) Terris-Prestholt et al.: The ‘No screening program ; ICER reported as not applicable’ is redundant, as this is the definition of the counterfactual/comparator, ICERs are ex pressed against this counterfactual/comparator.

RESPONSE: The redundant statement has now been removed. 

5) Kuznik et al: ICER is for what? Per infection averted? Please write that out. And clarify, what does an ICER mean for the ‘No screening program’, what was the comparator or counterfactual scenario here?

RESPONSE: This has now been clarified in Table 3 with a new column called ‘efficiency measures’ (column 9, row 10).

6) Rydzak: To interpret the ‘RST dominates’, please write out what is the comparator scenario for each scenario (No screening; RPR, and RST), for example: No screening compared to … RPR compared to No screening RST compared to ?RPR? or ?No screening?

RESPONSE: Table 3 has been revised and comparators for all full economic evaluations are detailed in column 5 of Table 3, while efficiency measures are now illustrated in column 10 of Table 3.

7) Schackman et al. Please clarify how Rural and Urban differed, e.g. in prevalence? Without that info, the differential results are meaningless to the reader.

RESPONSE: This has now been clarified in column 11, row 12 of Table 3. 

8) Vickerman et al: From the 4 outcomes (cost/DALY) given, RPR has lower cost/DALY for Serum but RST has lower cost/DALY for Whole Blood. Therefore, the conclusion ‘test using serum/whole blood: Cost-effective outcome = RST’ is not obvious from the results – the results suggest RST as more cost-effective for whole blood but not for serum. Please clarify; for example, add into the Results cell, the corresponding cost per DALY for Serum + Whole Blood combined.

RESPONSE: The key result for this study has now been clarified in column 11, row 13 of Table 3.

9) Blandford: As for Kuznik et al., above, please clarify the respective comparators for each of the 3 testing options. Also clarify ‘Not published but states ‘Dominiated’.

To facilitates readers’ matching between Conclusion and Results, I’d rephrase the Conclusion as ‘Off-site RPR and confirmatory TPHA’. And remove ‘(averts more cases)’ as that’s obvious from, and just a repeat of, the Results.

RESPONSE: Comparators for this study have now been added in column 5 and key study findings have been amended as requested and included in column 11.

10) Romoren et al: From the results, between the 4 options, the most cost effective looks to be Syndromic Management with azithromycin, at USD 21. In spite of that result, the conclusion states POC with azithromycin, despite it costing USD31 per cured case. Please reconciliate and clarify.

RESPONSE: This has now been clarified in column 11, row 17of Table 3. The results and the conclusion are now consistent. 

11) Larson et al.: Where does the 62% come from, what algorithm or policy was this the result of?

At reading the conclusion, I infer back that ‘RST’ must correspond to test and treat all, i.e. the third of the 3 options. Please write this out in the Results. And perhaps shorten the corresponding scenario names in column Intervention versus Comparator.

RESPONSE: The source of the 62% has now been added Footnote (denoted **) of Table 3). The link between the results and comparators have also been made clearer. The scenario names have also been made clearer. We have listed an explanation in the footnotes of Table 3 (denoted with **). 

Discussion:

The Discussion mentions many wise and true things, but they are not too well structured, leaving the reader without a clear understanding of the overall current evidence or its policy implications.

We would recommend structuring the discussion of limitations between:

1) Limitations that affect the core outcomes of studies analyzed (in Table 3), i..e. that bias those comparisons, notably:

• Societal perspective, which may cause under-stating the value POC compared to laboratory-based policies, if the savings from POC are for a large part on the patient/client side.

• Types of cost: if the variability across studies is important, then please show it – adding a column with ‘Cost components included’ to Table 3. (To make space in Table 3, you might remove the column Intervention versus Comparator’ which gets repeated in the next-right column in any case.?)

2) More general limitations or qualifications such as:

• Most studies costed only test kits, treatment and staff, but not QAQC or procurement: if this applies across all scenarios compared within a study, that comparison is not biased.

• Affordability and equity: True but somewhat beyond the scope of this review?

• Short time horizons: A true and important limitation – but not one that biases the comparison across scenarios within a study: Per treatment or maternal infections cured, the longer-term effects should be similar regardless of through which screening algorithm that treatment or cure was produced, not?

RESPONSE: Thank you for your comment, the limitations have now been grouped as recommended (see page 28, lines #370-383).

Conclusion:

The start and end of the conclusion as written are more an Introduction. I suggest to narrow and focus the conclusion on the part in the middle, from ‘Our review indicates…’ to ‘… other screening practices’.

RESPONSE: Thank you for your comment, we have made changes to the Conclusion as recommended by the reviewer.

Regarding the lack of geographical representation of the studies:

Why is that a problem? Is there reason to think that the alternative screening algorithms would compare differently in different settings (e.g. in Latin America or Asia-Pacific)? Why, what setting-specific determinant (e.g. STI prevalence??) would be the driver of varying cost-effectiveness rankings? If you believe this is truly important, then make it more concrete, beyond a description of geographic representation, in the 3rd paragraph of the Discussion. 

RESPONSE: A sentence has been added to the discussion emphasising the importance of conducting economic evaluations in a range of settings (see page 25, lines #292-295). 

Reviewer #2:

Presentation, tables and figures

• Readers should be able to understand the study objective in each of the figure and table titles. They should stand alone.

RESPONSE: Thank you for your suggestion, we have renamed all the tables and the figures in this manuscript to reflect the study objective in each.

References

• Reference 8 is old. There are recent papers on the global burden of curable STIs

• Your only reference for adverse outcomes is for syphilis. Add in evidence for other STIs too. 

• I think reference 18 should actually be 19. 

RESPONSE: Thank you for your feedback; we have included recent references for chlamydia, gonorrhoea, trichomonas, and bacterial infection: REFS #1-REF#9 (see page 3, line #59). We have replaced reference #8 with a newer study: REF#14 (see page 3, line #62). We have also replaced reference #18 with two newer references: REF#26 and REF#27 (see page 3, line #75). 

Abstract, Introduction, Methods, Results and Discussion

Abstract

Line 24: “tests” and “facilitates” don’t agree (should be testing and facilitates or tests and facilitate). 

RESPONSE: This sentence has now been edited (see page 1, line #24).

Line 26: You mention “efficiency” often, but do you really mean effectiveness? What are your efficiency measures?

RESPONSE: Efficiency measures are now listed for all studies in column 10 of Table 3. 

Line 37: should be “assessed”.

RESPONSE: This correction has been made (see page 1, line #38). 

Introduction

Paragraph 2 covers many ideas including: effectiveness of early detection and treatment of HIV and syphilis; interventions for managing STIs; syndromic management for CT/NG/TV; and lab-based diagnosis out of reach. I suggest breaking this paragraph up to have one paragraph on HIV & syphilis testing guidelines/protocols (e.g. types of testing conducted), and effectiveness; and another paragraph dedicated to the infections managed by syndromic management, problems with syndromic management and barriers to testing.

RESPONSE: Thank you for your suggestion, we have amended the manuscript to reflect this suggestion (see page 3-4, lines #64-70 and lines #71-78).

Paragraph 3 could benefit from the more description of point-of-care testing, such as desirable attributes (accuracy, minimal training, rapid, low cost, portability), etc. 

RESPONSE: Thank you for your suggestion. Desirable attributes for both serological and molecular point-of-care tests are described on page4 lines #83-85.

Minor 

Line 60: When referring to the bacteria/protozoa for the first time, it would be good to spell out (e.g. Chlamydia trachomatis)

RESPONSE: Indeed, when referring to the causative organism (bacteria/protozoa) it is convention to list the name of the bacteria in full. However, for this manuscript, we are referring to, and identifying, the infection/disease (chlamydia) rather than the bacteria (chlamydia trachomatis), therefore we have listed the infection in the manuscript.

Line 85: All countries need to prioritize investment, but LMIC may have tighter budget constraints.

RESPONSE: This edit has been made on page 5 (see lines #94- #95). 

Line 89: Is there a citation for the new landscape that is emerging? What do you mean by this?

RESPONSE: “…new landscape emerging” refers to the recently developed and effective technology for testing and treatment at point-of-care. This has been clarified on page 5 (line 98) by the inclusion of supporting references for Chlamydia, Gonorrhoea, and HPV (REFS #37, #41, #44). 

Lines 100-105: Are you not also comparing and contrasting effectiveness measures?

RESPONSE: Thank you for your feedback, the purpose of this review is to systematically review the evidence on economic evaluations of point-of-care testing to detect STIs in pregnancy. We are comparing and contrasting the primary outcome of these studies, which include costs and cost-effectiveness, not effectiveness measures per se and clarified this (see page 6, lines #111-112).

Methods

With respect to the literature review, I have several questions:

1) “Partial/ full economic evaluation” should be defined in the methods.

RESPONSE: Brief definitions of partial and full economic evaluations have been included in the methods section (see page 7, lines #139-144).

2) What software was used for the screening (e.g. Covidence)?

RESPONSE: 

We have revised the manuscript to include the software package (Microsoft Excel (version 365)) used for screening records, titles and abstracts (see page 7 line #135).

3) I have found that a lot of economic analysis has been conducted in the grey literature (e.g. by USAID/Health Policy Project, World Bank), why did you decide to exclude grey lit?

RESPONSE:

We excluded grey literature as it is may not always follow recommended guidelines for conducting economic evaluations and/or is not often peer reviewed. By excluding grey literature, we appreciate that we may have excluded some relevant evidence and acknowledge this as a limitation (see page 28, lines #379-381).

4) Does your search account for different spellings of diseases? (e.g. gonorrhea/ gonorrhoea/gonococcal)? What about prenatal (in addition to antenatal)?

RESPONSE:

The search terms were finalised in consultations with a medical librarian. The terms and syntax used (Table 2) account for different spelling. For example, “GONORRHEA/” takes the American spelling of Gonorrhoea (i.e gonorrhea) into consideration and different derivations of “Gonorrhoea”. Prenatal is the parent term for antenatal, however upon closer inspection during the development of the search strategy, the exclusion of prenatal did not influence the outcome of the search.

5) Was the third reviewer/resolver more senior or have more expertise in the subject area?

RESPONSE:

Yes, the third reviewer is more senior, and is a more experienced economist specialising in economic evaluations in low- and middle- income countries (LMIC) and has published 3 other systematic reviews in high impact journals including Lancet Infectious Diseases and Health Economics. 

With respect to the data abstraction process, I have several questions:

6) Did you abstract data associated with your quality appraisals (e.g. perspective, exchange rate)? Do you have a table that could be published in the supplemental documents?

RESPONSE: We have not abstracted the data for the quality appraisals. However, in response to the comments from reviewer 1 we amended Table 3, which now presents relevant details extracted from the studies included in this review and somewhat overlaps with the checklist items in the quality appraisal. We have also amended the manuscript to reflect this (see page 7, lines #152-156).

7) What software was used for abstraction (e.g. Qualtrics/excel)?

RESPONSE: Microsoft Excel (version 365) was used for abstraction. The manuscript has been revised to make this clearer (see page 7, line #152).

Results

With respect to Table 3, I have several questions/comments:

1) I think it’s important to include the type of cost assessed in each study. As you note later, many studies did not include capital costs. Capital costs associated with POC testing often make this strategy unaffordable (compared to lab testing) because an assay must be placed at each clinical site.

RESPONSE: Types of costs (Cost components) are now detailed in column #7 of Table 3.

2) It would be helpful to also add a column for effectiveness outcomes. If there’s no room, at least define what health outcomes are reflected in the DALYs (e.g. PID, preterm birth, conjunctivitis, etc.)

RESPONSE: In Table 3, the column entitled ‘efficiency measures’ incudes the primary measure of effectiveness used in the economic evaluation (see column #9). 

With respect to the results narrative, I have several questions/comments:

1) Please include more discussion on the details and quality of the outcomes included in the CEA/CUAs and separate out by syphilis/HIV and CT. Did authors use results from RCTs or meta-analyses? Were the effectiveness measures derived from an appropriate population – or are they all upper income populations? Was the intervention appropriate/similar? Do authors assume treated and cured infections have the same probability of an adverse outcome as never being infected?

RESPONSE: we have amended the text to highlight the number of studies conducted using trial data and those conducted using existing literature (See page 18, lines #200-201). 

2) In the quality assessment, do you consider how costs were collected (e.g. micro-costing / bottom up costing)?

RESPONSE: Thank you for your comment, we did not include this in our analysis. However, we know that only four studies (all of the partial economic evaluations) clearly stated that they utilised an ingredients-based approach, three of which also indicated utilising a step-down approach to costing. All of the full economic evaluations (n=12) did not specify a method for how costs were collected, we could only establish that these studies took an ingredients based approach, however, could not establish if another approach was also utilised. 

Line 226: Does “treatment coverage” include loss to follow-up? I would think loss to follow-up would be important when comparing PoC with lab-based testing.

RESPONSE: The key drivers presented in this sub-section are those that are identified by the studies included in this review, we have only collated them to compare/contrast them. Treatment coverage refers to the number of antenatal clinic attendees treated for an STI. However, we cannot extrapolate whether treatment coverage includes loss to follow-up (unless specifically stated in each study).

Discussion

1) The first paragraph isn’t really about your study. I would include this content in the introduction and use the first paragraph to summarize your most important findings. 

RESPONSE: Thank you for your suggestion, we have amended the first paragraph of the discussion as suggested (See page 25, lines #267- 282)

2) Discuss drivers of costs/cost-effectiveness

RESPONSE: Thank you for your comment, we have revised the manuscript to include a paragraph on the drivers of cost and cost-effectiveness (See page 26, lines #318- 325).

3) Discuss the outcomes that were included. The effectiveness of CT/NG/TV infection testing and treatment in terms of preventing adverse health outcomes (besides PID) is very limited. This may be another important reason why countries are hesitant to introduce expensive testing.

RESPONSE: Thank you for pointing this out, indeed there is limited evidence demonstrating the effectiveness of testing and treating CT/TV/NG in pregnancy to reduce adverse pregnancy and birth outcomes such as miscarriage, still births, neonatal deaths etc. However, as detection of CT/TV/NG has improved over the last decade, there has been a growing number of random control trials investigating the effectiveness of detecting CT/TV/NG in pregnancy to reduce adverse pregnancy and birth outcomes, enabling the expansion of evidence and providing a definitive conclusion whether testing and treating STIs in pregnancy reduces adverse pregnancy and birth outcomes (See page 26, lines #296-303).

4) In the conclusion you mention a gap in terms of measuring the “life time costs and effects of testing and treatment on mothers and their babies,” but you don’t mention this in the discussion. What’s available in terms of lifetime costs of adverse outcomes? Do we need more research here?

RESPONSE: In the discussion we mention that most economic evaluations are based on a relatively short time-horizon and overlook these long-term costs and effects (see page 26 lines #313- #317).

5) Are the results similar to economic analyses in high income countries (e.g. Ong and Ditkowsky in Australia and the US)?

RESPONSE: Cost-effectiveness studies for STIs, including that of Ong and Ditkowsky, conducted in high income countries tend to not focus on testing at point-of-care unless conducted in a remote community. Thus making it difficult to compare results of studies based in high-income countries.

Conclusion

1) I disagree that your review “highlights the role that economic evidence plays in moving the global health agenda beyond the utility of antenatal point-of-care testing and treatment for STIs towards incorporating resource allocation, which is a major determinant of implementation and scale-up.” I think your review identifies and summarizes the literature, but more research is needed to move the agenda forward.

RESPONSE: We have toned down this statement to these concerns (see pages 28-29). 

2) Again, mention cost/cost-effectiveness drivers.

RESPONSE: We have revised the manuscript to include the ‘drivers of cost and cost effectiveness’ in the conclusion (see page 29, lines #388-391).

3) Mention life time costs in the discussion if you are going to include in the conclusion

RESPONSE: Please see previous response. These costs are now mentioned in the discussion (see page 26 lines #313- #317).

---

## [Decision Letter · Decision Letter 1]

26 Feb 2021

PONE-D-20-31634R1

Economic evaluation of point-of-care testing and treatment for sexually transmitted and genital infections in pregnancy in low- and middle-income countries: A systematic review

PLOS ONE

Dear Dr. Saweri,

Thank you for submitting your manuscript to PLOS ONE. After careful consideration, we feel that it has merit but does not fully meet PLOS ONE’s publication criteria as it currently stands. Therefore, we invite you to submit a revised version of the manuscript that addresses the points raised during the review process.

We look forward to receiving your revised manuscript.

Kind regards,

Remco PH Peters, MD, PhD, DLSHTM

Academic Editor

PLOS ONE

Journal Requirements:

Reviewers' comments:

Reviewer's Responses to Questions

**Comments to the Author**

1. If the authors have adequately addressed your comments raised in a previous round of review and you feel that this manuscript is now acceptable for publication, you may indicate that here to bypass the “Comments to the Author” section, enter your conflict of interest statement in the “Confidential to Editor” section, and submit your "Accept" recommendation.

Reviewer #1: (No Response)

Reviewer #2: All comments have been addressed

2. Is the manuscript technically sound, and do the data support the conclusions?

Reviewer #1: Yes

Reviewer #2: Yes

3. Has the statistical analysis been performed appropriately and rigorously? 

Reviewer #1: N/A

Reviewer #2: Yes

4. Have the authors made all data underlying the findings in their manuscript fully available?

Reviewer #1: Yes

Reviewer #2: Yes

5. Is the manuscript presented in an intelligible fashion and written in standard English?

Reviewer #1: Yes

Reviewer #2: Yes

6. Review Comments to the Author

Reviewer #1: GENERAL/MAJOR COMMENTS

The authors have made revisions in response to each comment, which has strengthened the Methods, Results and Discussion – but I wonder if some of the changes are adequate, and some may actually introduce confusion or misinterpretation.

Table 3 is more readable now. Still, this being the key results, it merits further improvement – and an attempt to restore some meaningful content dropped since the original submission.

SPECIFIC COMMENTS

RESULTS, Table 3:

Drivers of cost and cost-effectiveness: This is a useful addition, in principle – but does not really yet add insight in practice, because:

• The drivers mentioned are much broader than those that determine the ranking between study arms, i.e. RST versus comparators, shown in the left-hand column.

• The drivers mentioned pertain mostly to the absolute cost per outcome (e.g. $$ per case averted or per DALY) but that column got dropped since the original submission.

• The cost drivers mentioned are named too broad and vague, for example: ‘Cost’ does this mean procurement cost i.e., test price?

I would suggest to be more concrete and specific in listing cost drivers -- focusing on the specific factors that explain the comparison between RST and comparators – and explain the direction of effect – which study arm/comparator had higher or lower value for each cost driver. Or optionally, slit the column into 2:

• Drivers of ‘absolute’ cost per outcome;

• Drivers of the difference in cost-per-outcome between RST and comparators.

You could make space by removing the column Infection studied, replacing that with one start or footnote indicating the one study where this was not syphilis but chlamydia. And some other of the left-hand columns might be merged.

For example, in Shelley et al., which of the 5 factors explained that RST rollout was cheaper than pilot? Which of the two used which type of blood collection? Which of the two arms had longer or shorter lifetime, and how did that influence the cost?

Sweeney et al.

• If the efficiency measure was total cost per health facility, did each health facility in both study arms test and treat equally many persons?

• What is meant with Time taken to test, is that the personnel time per test? Was that time longer for RST than for RPR?

Levin: In the column cost drivers, could you state the factors that explained the contrasting cost ranking between Mozambique and Bolivia? That does not seem to require the study having included a sensitivity analysis (?)

Owusu-Edusei et al: Was cost associated with adverse pregnancy outcome really a cost driver? Isn’t that relevant only if the incidence of such outcomes differed between the study arms i.e. comparators – which is unlikely: any undetected and untreated infection will on average incur the same adverse outcomes, regardless of the study arm /comparator that failed it? This is an example of my general comment above, that the many items mentioned in this column appear mostly meaningless. I suggest you limit the contents of this column to factors that influence the comparison between study arms / comparators. Any factors that influence the overall cost are not relevant, since your table (or any other Result in the paper) do not deal with the absolute overall cost or cost-effectiveness measures – they all just center on relative comparison of RST against comparators.

& Same comment for Rydzak.

Similarly, for Blandford, did the relative distribution of active/latent syphilis differ between the study arms i.e., comparators? If not, it is not relevant in this table that focuses on comparing test policies.

Kuznik 2015 & 2013, and Vickerman: Rephrase ‘All comparators’ as ‘Comparator’ since these studies each had only 1 comparator.

Terris-Prestholt et al.: As for Levin above, please focus and limit the cost drivers mentioned on those that explain the difference in ranking (RST above or below comparators) between Peru, and Tanzania+Zambia..

Kuznik 2013: By definition, the Discount rate will always influence the absolute cost measure, but it will do so equally for all study arms and comparators, and all studies not just Kuznik – so this is a meaningless redundant item to mention here.

Rydzak, Shackman, & Romoren:: Rephrase ‘All comparators’ as ‘Both comparator’ since these studies each had 2 comparators.

Rydzak: 1000 women is women tested, or treated?

Romoren et al: Under point 3. Remove the redundant words ‘Point-of-care testing and Azithromycin treatment vs.’

In the right-most cell of this row, clarify that PoC is RST, and replace ‘and’ by ‘with’ to clarify that this (combo of test type + treatment type) is 1 study arm not 2.

Larson: The added footnote (answering my earlier comment: Where does the 62% come from, what algorithm or policy was this the result of?) is helpful. But I’m still left with the question, why only 10% of positive cases were treated, and which 10% was this, was this random or by any characteristic or policy criteria?

& Cosmetic/formatting: Make Table 3 shorter and more readable by adapting the column widths according to their average content.

DISCUSSION

Page 36: Please explain the ‘Largest vs smallest percentage difference, 170% and 19% - between what? For which outcome is this, and which 4 of the studies listed in Table 3 provided the extremes?

Page 37: ‘In particular, the fundamentals of an economy and disease epidemiology..’: This reads as poor-language generality, I don’t understand what specifically the author want to say. Please rephrase into something more concrete, or remove.

‘This has, in turn, led to a comparatively new and emerging field…’: I think the discussion could do without this perhaps exaggerated statement.

‘Economic evaluations… should ideally… incorporate the long-term costs and benefits…’: I am not convinced this is necessarily neede -- unless the longer-term costs and/or benefits differ between the intervention arms being compared? Which does not seem to be the case for POC STI tests. Please also see my related specific comments on various rows/study summaries in Table 3.

Page 40: ‘… the incorporation of extended cost-effectiveness analyses and budget impact analyses into cost-effective analyses…: This reads like triple circular.

Additionally, most studies took a narrow view… short-run time horizon, although this does not influence the comparability between studies, it does highlight a short/coming of the studies included in this review. Consider to shorten and simplify – the time horizon was discussed already earlier in the Discussion.

‘…therefore grey literature…’: These last 10 last words are redundant in the sentence.

‘Finally, a further enhancement … equity’: This duplicates the mentioning of equity in line 403 above?

Reviewer #2: The manuscript is improved and my comments have been addressed. My suggested minor revision is related to the percentage difference mentioned in the results and discussion. Please mention the percentage difference (and how it was calculated) as an outcome of interest in the methods.

7. PLOS authors have the option to publish the peer review history of their article (what does this mean?). If published, this will include your full peer review and any attached files.

Reviewer #1: **Yes: **Eline Korenromp

Reviewer #2: **Yes: **Adriane Wynn

---

## [Author Response · Author response to Decision Letter 1]

12 Apr 2021

Editor-in-Chief

Plos one

Manuscript Title: Economic evaluation of point-of-care testing and treatment for sexually transmitted and genital infections in pregnancy in low- and middle-income countries: A systematic review

Authors: Olga PM Saweri, Neha Batura, Rabiah al Adawiyah, Louise Causer, William Pomat, Andrew J Vallely and Virginia Wiseman

On behalf of my co-authors, I would like to thank the editor and reviewers for providing a detailed and extensive second review of our manuscript. We greatly appreciate the time and effort put into the feedback we have received. We have revised the manuscript and our responses to the reviewers’ feedback are provided below. 

Best, 

Olga Saweri

RESULTS, Table 3: 

1. Drivers of cost and cost-effectiveness: This is a useful addition, in principle – but does not really yet add insight in practice, because:

• The drivers mentioned are much broader than those that determine the ranking between study arms, i.e. RST versus comparators, shown in the left-hand column.

• The drivers mentioned pertain mostly to the absolute cost per outcome (e.g. $$ per case averted or per DALY) but that column got dropped since the original submission.

• The cost drivers mentioned are named too broad and vague, for example: ‘Cost’ does this mean procurement cost i.e., test price?

RESPONSE:

Thank you for your feedback. The results presented in Tables 3 and 4 illustrate the results published in each included study and our interpretation of those results. The drivers of cost and cost-effectiveness, which were previously included in Table 3 are now presented in Table 4; these drivers are the results of sensitivity analyses conducted by each study included in this review (see Column D in Table 4 on pages 16-22). The revisions made to Column D in Table 4 (pages 16-22) reflect the results published by each study included in this review and are less ambiguous. In addition, we specifically list the drivers of cost and cost-effectiveness in the manuscript text and cross-reference the results presented in Table 4 (see page 24, lines 248-252). 

2. I would suggest to be more concrete and specific in listing cost drivers -- focusing on the specific factors that explain the comparison between RST and comparators – and explain the direction of effect – which study arm/comparator had higher or lower value for each cost driver. Or optionally, slit the column into 2:

• Drivers of ‘absolute’ cost per outcome;

• Drivers of the difference in cost-per-outcome between RST and comparators.

RESPONSE:

Thank you for this suggestion. We have revised the manuscript to reflect this wherever possible. However, this was challenging. Few studies explicitly differentiated between drivers of absolute cost per outcome and those the explain the comparison between study arms or comparators. Most studies do not explicitly state that the results of the sensitivity analysis were drivers of cost-effectiveness, cost, effectiveness or between arms (see Table 4, column D, pages 18-24). Where the text is not explicit, we have assumed that the results of the sensitivity analysis are drivers of cost-effectiveness for full economic evaluations (n=12) or drivers of cost for partial economic evaluations (n=4).

3. You could make space by removing the column Infection studied, replacing that with one start or footnote indicating the one study where this was not syphilis but chlamydia. And some other of the left-hand columns might be merged.

RESPONSE:

We have amended Table 3 and included an additional table (Table 4) to improve the spaces in each column heading to improve readability (see Table 3 on pages 10-15 and Table 4 on pages 16-22).

4. For example, in Shelley et al., which of the 5 factors explained that RST rollout was cheaper than pilot? Which of the two used which type of blood collection? Which of the two arms had longer or shorter lifetime, and how did that influence the cost?

RESPONSE:

In this paper, the sensitivity analysis parameters were screening coverage, test kit price, supply wastage, project life, and type of blood collection. Blood collection here refers to the difference in collecting blood via a finger prick and via venipuncture. The former is utilized for RST while the latter for RPR. The results of the sensitivity analysis indicate that cost per person screened is most sensitive to changes in screening coverage, however changes in RST test kit and supply wastage also impact cost per person. These results are now reflected in table 4 column D (pp18-24).

In response to the query about the time horizon, the costs of both the pilot and the rollout were collected over the same amount of time, just two years apart. Further, the sensitivity analysis measured the effect of the project life, whereby costs in the rollout period were more sensitive to a change in the project life compared to the pilot period.

5. Sweeney et al. 

• If the efficiency measure was total cost per health facility, did each health facility in both study arms test and treat equally many persons?

• What is meant with Time taken to test, is that the personnel time per test? Was that time longer for RST than for RPR?

RESPONSE

No, in this study, health facilities did not test and treat equal numbers of women in the study arms. In the RPR arm, over a 9-month period between 2007 and 2008, 838 antenatal clinic (ANC) attendees were tested for syphilis at 6 of the 9 selected health facilities. This is 17.8% of ANC attendees. Further 27% of women tested positive for syphilis and about 66% of those who tested positive were treated. The results state that no confirmatory tests were completed, therefore the true prevalence is unknown.

In the RST arm, over a similar time frame (in 2009-2010) at the 9 selected health facilities, 9372 ANC attendees were tested (87% of total ANC attendees); 10% of antenatal clinic attendees tested positive, of which 92% were treated.

With respect to ‘the time taken to test’, there is a difference between the time taken to conduct an Rapid Plasma Reagin (RPR) test for syphilis compared to a point-of-care test, or rapid syphilis test (RST). An RST is ‘quicker’, the study does not specifically state the difference in time. The sensitivity analysis showed that both RST and RPR costs were sensitive to staff time. 

6. Levin: In the column cost drivers, could you state the factors that explained the contrasting cost ranking between Mozambique and Bolivia? That does not seem to require the study having included a sensitivity analysis (?)

RESPONSE: It is not clear from Levin et al (2007), therefore we have not amended the manuscript. What is clear from Levin et al (2007) is that direct comparisons between total costs in each country would not lead to “meaningful” results as the ‘project periods and the numbers of women screened are not directly comparable’ (see page S50/S54). 

With respect to the rank of costs, costs related to testing accounted for 50% to 84% of costs in both countries. Looking into the breakdown of costs, ‘tests and other supplies’ accounted for the greatest share of costs in both countries. In Bolivia, clinic staff salaries associated with testing accounted for the second largest share of costs followed by start-up costs. However, in Mozambique, lab supplies accounted for the second largest share of costs followed by clinic staff salaries. 

7. Owusu-Edusei et al: Was cost associated with adverse pregnancy outcome really a cost driver? Isn’t that relevant only if the incidence of such outcomes differed between the study arms i.e. comparators – which is unlikely: any undetected and untreated infection will on average incur the same adverse outcomes, regardless of the study arm /comparator that failed it? This is an example of my general comment above, that the many items mentioned in this column appear mostly meaningless. I suggest you limit the contents of this column to factors that influence the comparison between study arms / comparators. Any factors that influence the overall cost are not relevant, since your table (or any other Result in the paper) do not deal with the absolute overall cost or cost-effectiveness measures – they all just center on relative comparison of RST against comparators. 

RESPONSE:

Thank you for your comment. Yes, cost associated with adverse pregnancy outcome was a driver of cost, thus we have not made any amendments to the manuscript. Owusu-Edusei et al (2011) identifies drivers in the text, which we have included in this table. Table 2 (see page 1001/1003) illustrates that the incidence of adverse outcomes differs per study arm (total adverse pregnancy outcomes are highest when there is no screening program (n=39) and lowest when screening using an RST (n=2)).

Excerpt from Owusu-Edusei et al (2011) p1000/1003: “…we varied select variables and examined the total expected cost keeping all other variables constant. As expected, the relative test performance, test costs, and costs associated with adverse pregnancy outcomes were all influential.” The text continues to describe how all scenarios were cost-saving and the extent to which they were. 

8. Same comment for Rydzak.

RESPONSE:

Thank you for your feedback, an excerpt from Rydzak and Goldie (2008) page 780/784 indicates which variables were drivers, including the ‘costs associated with congenital syphilis’, therefore we have not made this change to the manuscript. The excerpt: “variations in test performance, test costs, and costs associated with congenital syphilis had the greatest effect on results. The choice between the 2 single-visit strategies was influenced by their comparative test sensitivity, and the cost of the ICS [rapid syphilis test] test kit, labour and supplies. A screening strategy was considered the preferred strategy if it achieved greater benefits at lower cost than all other strategies.”

Further, table 3 (page 779/784) indicates the number of adverse pregnancy outcomes and the number of adverse outcomes averted under each comparator. The table demonstrates that adverse pregnancy outcomes differ per comparator. 

9. Similarly, for Blandford, did the relative distribution of active/latent syphilis differ between the study arms i.e., comparators? If not, it is not relevant in this table that focuses on comparing test policies.

RESPONSE: 

Thank you for your comment; Blandford et al (2007) illustrates on that ‘model effectiveness of antenatal screening approaches are most affected by test sensitivity’ (see page S64/S66). The article also states that prevalence of maternal syphilis and the distribution of active and latent syphilis indirectly impact cost-effectiveness through costs. Given this, we have not changed the drivers stated in Table 4 (see page 20). 

10. Kuznik 2015 & 2013, and Vickerman: Rephrase ‘All comparators’ as ‘Comparator’ since these studies each had only 1 comparator.

RESPONSE:

Thank you for your suggestion. We have made relevant changes used consistent language across all studies. The key findings column has now been placed in Table 4 (see Column E, Table 4 on pages 16-22).

11. Terris-Prestholt et al.: As for Levin above, please focus and limit the cost drivers mentioned on those that explain the difference in ranking (RST above or below comparators) between Peru, and Tanzania+Zambia.

RESPONSE:

Thank you for your comment, Terris-Prestholt er al (2015) states on page S76/S80 that the ‘variability in the DALYs averted for this algorithm is due to the variability of RST reactivity rate (or positivity rate)’ in addition, the study also highlights that variability in total cost is driven primarily by ANC attendance, fixed clinic costs and screening coverage. These have all been included as drivers of cost and cost-effectiveness. We have not included ‘true prevalence’, as this did not affect the ranking (See Row 11, Column D in Table 4 on page 19)

12. Kuznik 2013: By definition, the Discount rate will always influence the absolute cost measure, but it will do so equally for all study arms and comparators, and all studies not just Kuznik – so this is a meaningless redundant item to mention here.

RESPONSE:

Thank you for your feedback, we have amended Table 4 to reflect this (see Column D, Table 4 on pages 16-22).

13. Rydzak, Shackman, & Romoren:: Rephrase ‘All comparators’ as ‘Both comparator’ since these studies each had 2 comparators.

RESPONSE:

Thank you for your suggestion. We have made relevant changes to Table 4 and used consistent language across all studies (see Column D, Table 4 on pages 16-22).

14. Rydzak: 1000 women is women tested, or treated?

RESPONSE:

Rydzak and Goldie et al (2008) states that their outcome measure is ‘discounted costs saved per 1000 women’, which includes both testing and treatment for syphilis.

15. Romoren et al: Under point 3. Remove the redundant words ‘Point-of-care testing and Azithromycin treatment vs.’ In the right-most cell of this row, clarify that PoC is RST, and replace ‘and’ by ‘with’ to clarify that this (combo of test type + treatment type) is 1 study arm not 2.

RESPONSE:

Thank you for your comment. We have amended the manuscript to reflect that ‘testing for chlamydia using a point-of-care test followed by treating those testing positives for chlamydia with azithromycin’ was the most cost-effective outcome (see Row 18, Column E, Table 3 on pages 14 and Row 18, Column D, Table 4 on page 21).

We would like to clarify that the point-of-care test here refers to a point-of-care test for chlamydia, and therefore is not a rapid syphilis test (RST). We have amended Table 3 and 4 to state ‘point-of-care’ test.

16. Larson: The added footnote (answering my earlier comment: Where does the 62% come from, what algorithm or policy was this the result of?) is helpful. But I’m still left with the question, why only 10% of positive cases were treated, and which 10% was this, was this random or by any characteristic or policy criteria?

RESPONSE:

We have edited the contents of table 3 to improve clarity (see Row 19, Column E of Table 3) and the footnote of Table 3 (see pages 14-15).

Larson et al (2014) built three scenarios using the results of an evaluation study exploring rollout of point-of-care testing and treatment for syphilis in antenatal clinics (ANC). The evaluation found that 62% of ANC attendees were tested for syphilis. All women testing positive should have been treated, but only 10% were. For clarity, the scenarios from this study are outlined below:

Scenario 1 (baseline): the results of the evaluation study found that 62% of attendees were tested for syphilis, while only 10% of those that tested positive were treated. Therefore scenario 1 emulated this: 62% of antenatal clinic attendees tested for syphilis and 10% of test positives were treated. 

Scenario 2: explored the scenario of treating all antenatal clinic attendees testing positive yet keeping the number of antenatal clinic attendees tested for syphilis (62%) constant. Thus scenario 2 becomes testing 62% of antenatal clinic attendees and treating all antenatal clinic attendees that test positive for syphilis.

Scenario 3: which could be seen as the ideal scenario, explores the possibility of testing all antenatal clinic attendees and treating all of whom test positive for syphilis.

17. Cosmetic/formatting: Make Table 3 shorter and more readable by adapting the column widths according to their average content.

RESPONSE:

Thank you for your comment. We have edited and adjusted Table 3 to make it more readable (pages 10-15).

DISCUSSION

1. Page 36: Please explain the ‘Largest vs smallest percentage difference, 170% and 19% - between what? For which outcome is this, and which 4 of the studies listed in Table 3 provided the extremes?

RESPONSE:

The percentage differences have been calculated per study to express the measured differences between comparators of each study included in this review. We have included percentage differences in Column G of Table 4 (see pages 16-22) to illustrate the difference between the baseline figure and the most cost-effective comparator of each study included in this review. In addition to percentage differences the cost-effectiveness decision rule has been included in Column C of Table 4 to reflect how each study determined cost-effectiveness. Together these findings illustrate between-study variability (see lines 285-293, page 30). In addition, we have edited the methods section to highlight percentage differences briefly and their inclusion in the results (see lines 163-168, pages 7-8).

2. Page 37: ‘In particular, the fundamentals of an economy and disease epidemiology..’: This reads as poor-language generality, I don’t understand what specifically the author want to say. Please rephrase into something more concrete, or remove.

RESPONSE:

Thank you for your feedback, we have amended the manuscript (see lines 307-311, pages 30-31). 

3. ‘This has, in turn, led to a comparatively new and emerging field…’: I think the discussion could do without this perhaps exaggerated statement.

RESPONSE: 

Thank you for your comment, we have amended the manuscript (see lines 313-316, page 31).

4. ‘Economic evaluations… should ideally… incorporate the long-term costs and benefits…’: I am not convinced this is necessarily neede -- unless the longer-term costs and/or benefits differ between the intervention arms being compared? Which does not seem to be the case for POC STI tests. Please also see my related specific comments on various rows/study summaries in Table 3. 

RESPONSE:

Thank you for your feedback, the effects of untreated sexually transmitted infections in pregnancy can extend after birth, both in women and in children. Estimates of long-term costs and benefits would help us understand whether the investment in such programs can have a positive impact on the health of mothers and babies later in life. 

5. Page 40: ‘… the incorporation of extended cost-effectiveness analyses and budget impact analyses into cost-effective analyses…: This reads like triple circular. 

RESPONSE:

Thank you for your suggestion, we have amended the manuscript to sound less ‘like [a] triple circular’ (see lines 379-382, page 33).

6. Additionally, most studies took a narrow view… short-run time horizon, although this does not influence the comparability between studies, it does highlight a short/coming of the studies included in this review. Consider to shorten and simplify – the time horizon was discussed already earlier in the Discussion. 

RESPONSE:

We have shortened the sentence as suggested (see lines 394-395, page 33). 

7. ‘…therefore grey literature…’: These last 10 last words are redundant in the sentence.

RESPONSE:

We have removed this sentence from the manuscript (see line 398, page 33).

8. ‘Finally, a further enhancement … equity’: This duplicates the mentioning of equity in line 403 above?

RESPONSE:

The previous round of reviewer comments suggested that the final paragraph of the discussion should be a ‘summary of the discussion’. Thus, the final sentence of this paragraph seems repetitious as it is a summary point of the discussion. We have amended the manuscript to sound less repetitious (see line 399-400, page 33).

---

## [Editor Report · Decision Letter 2]

31 May 2021

Economic evaluation of point-of-care testing and treatment for sexually transmitted and genital infections in pregnancy in low- and middle-income countries: A systematic review

PONE-D-20-31634R2

Dear Dr. Saweri,

We’re pleased to inform you that your manuscript has been judged scientifically suitable for publication and will be formally accepted for publication once it meets all outstanding technical requirements.

Kind regards,

Remco PH Peters, MD, PhD, DLSHTM

Academic Editor

PLOS ONE
---

## [Editor Report · Acceptance letter]

9 Jun 2021

PONE-D-20-31634R2 

Economic evaluation of point-of-care testing and treatment for sexually transmitted and genital infections in pregnancy in low- and middle-income countries: A systematic review 

Dear Dr. Saweri:

I'm pleased to inform you that your manuscript has been deemed suitable for publication in PLOS ONE. Congratulations! Your manuscript is now with our production department. 

Kind regards, 

on behalf of

Prof Remco PH Peters 

Academic Editor

PLOS ONE